# DISCOVERYWORLD: A Virtual Environment for Developing and Evaluating Automated Scientific Discovery Agents

**Peter Jansen**[*‡]**, Marc-Alexandre Côté**[†]**, Tushar Khot**[*] **Erin Bransom**[*]**, Bhavana Dalvi Mishra**[*]**,
Bodhisattwa Prasad Majumder**[*]**, Oyvind Tafjord**[*]**, Peter Clark**[*]
[*]Allen Institute for Artificial Intelligence    [†]Microsoft Research    [‡]University of Arizona
peterj@allenai.org

## Abstract

Automated scientific discovery promises to accelerate progress across scientific domains. However, developing and evaluating an AI agent's capacity for end-to-end scientific reasoning is challenging as running real-world experiments is often prohibitively expensive or infeasible. In this work we introduce DISCOVERYWORLD, the first virtual environment for developing and benchmarking an agent's ability to perform complete cycles of novel scientific discovery. DISCOVERYWORLD contains a variety of different challenges, covering topics as diverse as radioisotope dating, rocket science, and proteomics, to encourage development of *general* discovery skills rather than task-specific solutions. DISCOVERYWORLD itself is an inexpensive, simulated, text-based environment (with optional 2D visual overlay). It includes 120 different challenge tasks, spanning eight topics each with three levels of difficulty and several parametric variations. Each task requires an agent to form hypotheses, design and run experiments, analyze results, and act on conclusions. DISCOVERYWORLD further provides three automatic metrics for evaluating performance, based on (a) task completion, (b) task-relevant actions taken, and (c) the discovered explanatory knowledge. We find that strong baseline agents, that perform well in prior published environments, struggle on most DISCOVERYWORLD tasks, suggesting that DISCOVERYWORLD captures some of the novel challenges of discovery, and thus that DISCOVERYWORLD may help accelerate near-term development and assessment of scientific discovery competency in agents. Code available at github.com/allenai/discoveryworld.[1]

## 1 Introduction

A long-standing dream of AI has been to build systems that can perform scientific discovery, potentially leading to new breakthroughs for the benefit of humanity [13]. Recently, with the rise of neural techniques, there have been several successful discovery systems developed for specialized problems such as protein folding [10, 15], mathematics [22], and material science [24]. However, while the results have been impressive, these systems (deliberately) bypass the full discovery process of ideation, hypothesis formation, experiment design, etc., and instead (expertly) perform systematic searches over a pre-defined hypothesis space, with pre-defined goals. This raises the question: how much more can be achieved if AI is applied to the broader scientific process? Some works have indeed developed early systems for this, for example, in chemistry [1], and genetics [12]. These systems can also generate hypotheses, design experiments, and execute them (via robotics) in real environments. However, operating in real environments is expensive and complex, creating a barrier

---

[1]Released under Apache-2.0 license.

38th Conference on Neural Information Processing Systems (NeurIPS 2024) Track on Datasets and Benchmarks.

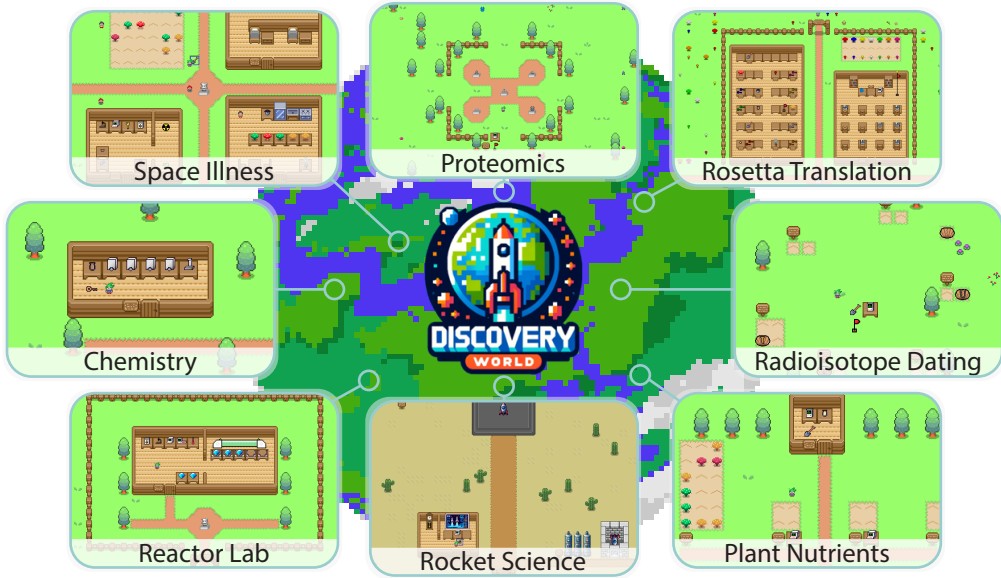

Figure 1: DISCOVERYWORLD is a virtual environment for developing and evaluating discovery agents, with challenge tasks covering a broad variety of different topics such as those shown above.

for entry. In addition, real environments inevitably encourage a focus on task-specific details, at the potential cost of developing more general discovery skills in an agent.

Our goal is to help remedy these by creating the first virtual discovery environment where solving tasks *demands all of the key facets in end-to-end scientific discovery*,[2] and which covers a *broad variety* of discovery topics. Our approach is to develop a text-based simulated world (with optional 2D visual overlay), called DISCOVERYWORLD, where agents can navigate around, interact with objects in the world, use scientific equipment (measuring devices, tools, etc.), and make observations. Agents can then form hypotheses, plan and execute experiments, and draw conclusions to solve challenge tasks developed for this virtual world. DISCOVERYWORLD tasks are grounded in eight varied topics, such as radioisotope dating, rocket science, and proteomics, to encourage development of agents with *general* discovery skills rather than hard-wiring to a particular challenge (see Figure 1). The tasks themselves are realistic (but simplified), allowing agents to apply both scientific and commonsense knowledge when attempting them. DISCOVERYWORLD thus provides an environment for exercising and evaluating general-purpose skills in end-to-end AI discovery systems (see Figure 2).

DISCOVERYWORLD is inspired by a growing number of text-based simulation environments [5, 31, inter alia], while also being novel in both its environment and tasks:

- DISCOVERYWORLD tasks are long-horizon, requiring multiple facets of discovery including ideation, experimentation, systematic search, and analysis to be performed to solve a task.
- The tasks do not suggest a solution approach, instead requiring the agent to ideate and define hypotheses to explore. This contrasts with tasks in many adventure game environments where solution approaches are often more stylistic or constrained.
- DISCOVERYWORLD is realistic (but simplified) rather than counterfactual, so that background knowledge can be sensibly applied.
- The tasks cover eight diverse topics, from identifying the cause of space illnesses to reactor tuning, to encourage development of general rather than task-specific solutions.

Finally, automatically evaluating an agent's progress on a discovery task is itself challenging. We devise a three-part evaluation strategy to help with this, more on this in Section 3.4.

Our contributions are thus:

---

[2]While a precise definition of "scientific discovery" is somewhat elusive [17, 19], we here adopt a pragmatic approach: Our goal is to help develop systems that can perform the end-to-end research process that human scientists engage in as they work on a problem, attempt to answer a question, or more generally advance their field of study. We use the term "scientific discovery" here to refer to this process.

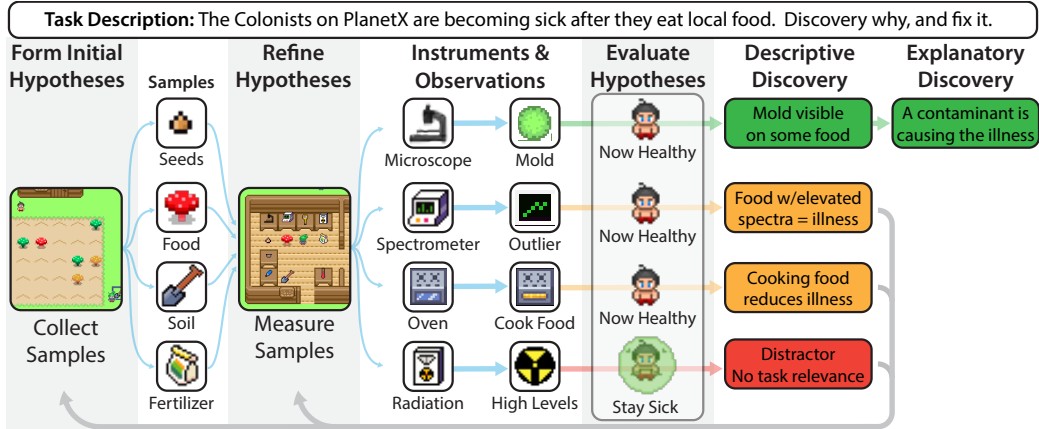

Figure 2: DISCOVERYWORLD tasks require end-to-end scientific discovery, from ideation, hypothesis formation, experiment design, data collection and analysis, forming conclusions, and acting on results. Distractors and task solutions that provide only descriptive discoveries require agents to frequently iterate hypotheses and experiments to reach full explanatory discoveries.

- We introduce the first virtual environment for benchmarking an agent's general ability to perform complete cycles of novel scientific discovery.
- A comprehensive evaluation set of 120 different tasks for this environment, spanning eight diverse topics, each with three levels of difficulty and several parametric variations.
- An evaluation framework allowing automatic evaluation of agents in DISCOVERYWORLD.
- Baseline results for agents in this environment, illustrating that DISCOVERYWORLD captures several novel challenges that contemporary agent models struggle with.

Together, we hope DISCOVERYWORLD will inspire and accelerate the development of new, general AI discovery agents by the community.

## 2 Related Work

Recently, several real-world discovery systems have shown success in areas such as genetics (Adam [12], Eve [32]), chemistry (CoScientist [1], ChemCrow [2]) and protenomics (AlphaFold [10], RoseTTAFold [15]). However, these systems are expensive and task-specific. Inspired by these, DISCOVERYWORLD aims to be a broad coverage, virtual environment allowing general scientific research skills to be developed and evaluated. There are many virtual environments that touch on aspects of the discovery process (see Table 1), although DISCOVERYWORLD is the first aimed at the full end-to-end pipeline:

Many **physical simulation environments** were developed (e.g., for robotics), focusing on object manipulation and navigation. Some are visual/spatial environments (e.g., AI2-Thor [14], ALFRED [25]), while others are textual/symbolic (e.g., TextWorld [5], MiniGrid [4], ALFWorld [26]).

Many **game worlds** require exploration and discovery, e.g., NetHack [16], MineDojo [6]. However, these operate in counterfactual worlds where good scientific choices are not always rewarded, and are primarily aimed at entertainment.

Some virtual environments (abstractly) cover **real world** tasks. WebArena [36] simulates Web-based activities, e.g., browsing for a phone-number, on-line shopping, aiming to improve an agent's task-specific Web navigation skills. Perhaps closest to DISCOVERYWORLD is ScienceWorld [31], a text-based environment for solving simple science quests known to elementary science students, such as "convert a liquid into a solid" (e.g., put water in the freezer). ScienceWorld requires commonsense object manipulation at the level of an elementary science student, but not ideation or systematic search to complete tasks. In contrast, we show that DISCOVERYWORLD contains discovery tasks that are challenging even for human scientists with PHDS in the natural sciences.

Finally, some environments are explicitly designed to host **hypothesis search**. Alchemy [28] is a 3D video game to find which mixture of potions transforms a stone into a more valuable form.

Table 1: A comparison of existing virtual environments with DISCOVERYWORLD. Note that to control for spatial complexity across environments, for the purposes of counting actions, move actions are considered single actions (i.e. *move [dir]*). Information in this table has been pieced together by looking at the different source materials for each environment (e.g., paper, website, and codebase). These should be taken as our best estimate.

| Environment | Multi-modal | Domain | # of Tasks | Para-metric | # Obj. Props. | # of Actions | Task Length |
|---|---|---|---|---|---|---|---|
| MINIGRID [4] | Image/Symbol | Pick+Place | 23 | Yes | 4 | 4 | 85 |
| ALFRED [25] | Image | Pick+Place | 6 | Yes | 20 | 7 | 50 |
| ALFWORLD [26] | Text/Image | Pick+Place | 6 | Yes | 16 | 9 | 10 |
| MINEDOJO [6] | Image | Minecraft | $10^\dagger$ | Yes | 256+ | 12 | 100k |
| NETHACK LE [16] | Text+Image | Dungeon | 1 | Yes | 69 | 78 | 80k |
| ALCHEMY [28] | Image/Symbol | Chemistry | 1 | Yes | 3 | 9 | 200 |
| IVRE [33] | Image/Symbol | Hypothesis Testing | 1 | Yes | 3 | 9 | 10 |
| SCIENCEWORLD [30] | Text | Elem. Science | 30 | Yes | 36 | 25 | 100 |
| **DISCOVERYWORLD** | Text+Image | Sci. Discovery | 24+10 | Yes | 63 | 14 | 1k |

Table 2: High-level descriptions of the 8 *discovery themes* in DISCOVERYWORLD, with full task descriptions (including spoilers) provided in APPENDIX B.3. It is from these 8 discovery themes $\times$ 3 difficulty levels that we parametrically generate 120 unique instances of discovery tasks.

| Theme | Description |
|---|---|
| Proteomics | Identify which species in a region migrated in the recent past by discovering that one is an outlier in a clustering analysis of protein concentration values. Higher difficulties involve more data dimensions. |
| Chemistry | Manufacture a rust removal agent by mixing different concentrations of chemicals then testing those solutions, guided by a hill-climbing signal of decreased rust that can reduce the search space. |
| Archaeology | Validate radioisotope dating by correlating radioisotope levels with known artifacts' ages, choosing the correct radioisotope between several alternatives for dating, then identify the oldest unknown artifact. |
| Reactor Lab | Discover a relationship (linear or quadratic) between a physical crystal property (like temperature or density) and its resonance frequency through regression, and use this to tune and activate a reactor. |
| Plant Nutrients | Discover that plants on PLANET X prefer specific combinations of nutrients that follow logical rules (e.g. XOR, AND, OR, NOT), then grow plants by setting soil nutrient levels that follow those rules. |
| Space Sick | Investigate the cause of a mild and occasional colonist illness in response to eating local food, then formulate and implement a solution so that future colonists no longer contend with this illness. |
| Rocket Science | Measure a number of unknown planetary properties (such as the radius and mass of PLANET X), then use provided equations to calculate orbital velocity and propellant needs for a rocket launch. |
| Translation | Explore an environment to infer the meanings of words in an unknown language by grounding them to observations of specific objects and actions, then take actions based on the translated utterances. |

Similarly in IVRE [33], users perform artificial category experiments to identify which blocks are "blickets". These environments exercise the systematic search part of the discovery pipeline. Likewise, MLAgentBench [8] requires software experiments to solve a challenge (improve a ML algorithm), but in the constrained environment of ML software. In contrast, DISCOVERYWORLD aims to cover a broad range of tasks in a (simulated) physical environment, covering the full end-to-end discovery pipeline.

## 3 DISCOVERYWORLD Simulator and Environments

### 3.1 Simulator

**Engine:** DISCOVERYWORLD is implemented in a custom simulation engine developed from the ground-up to enable building dynamic discovery simulations with a variety of object properties and simulated object behaviors. Every object in DISCOVERYWORLD is constructed from materials with measurable properties, many of which are observable with instruments available in the environment (a list of 60+ frequent object properties is provided in APPENDIX B.5). The simulator is implemented as approximately 20K lines of PYTHON using the PYGAME framework [21], and provides both an API for developing agents, as well as a graphical user interface for humans to play DISCOVERYWORLD tasks. The API resembles the OPENAI GYM specifications [3, 27], such that at each step, the agent

---

$^\dagger$We only considered programmatic tasks as they can be measured accurately. Then, we only count the (sub)categories since most programmatic tasks are parametric variations of those.

is provided with an observation from the environment, and must choose a single action to take during that turn from a set of possible actions. An automatic scorer runs in the background, and a given task continues until either it is solved, or the agent/human ends the simulation.

**Environment Space:** All current environments are represented as a $32 \times 32$ tile grid. As in text game simulators, all objects at a given tile are represented by an *object tree* [9], where the root node contains objects that are directly on the environment tile (such as a *table*), while child nodes of each object contain their contents (such as a *jar on the table*).

**Observations:** Observations in DISCOVERYWORLD can be provided as text, visual output, or both. Text observations are provided as JSON[3] and contain the following: (1) a list of all objects near the agent, their names, text descriptions, and unique identifier numbers; (2) a list of objects in the agent's inventory; (3) a list of objects the agent can directly interact with (either by being in the agents inventory, or directly beside the agent); (4) the agent's current world location, direction, and directions it can move to (that are not blocked by objects); (5) whether the agent is currently engaged in dialog with another agent, and if so, what pre-determined options it can say; (6) the current task description, and whether or not it has been completed. For tasks that requre interacting with other agents (such as *Space Sick*), we also implemented *DiscoveryFeed*, a Twitter-like posting environment that reduces the need for an agent to continually visit other agents to check on their status. The most recent *DiscoveryFeed* posts are included in the observation.

The visual observation contains a screenshot (768px$\times$512px) similar to the user interface, which provides a $24 \times 16$ tile view of the world (each tile is 32px$\times$32px) centered on the agent (see Figure 4 in the APPENDIX). This provides information about objects that are farther away than the JSON observation, which is typically limited to a certain distance (configurable, default within 3 tiles) around the agent due to the size (in tokens) of the objects and their associated descriptions.

**Action space:** DISCOVERYWORLD includes 14 possible actions, most of which are common actions such as *taking, dropping, or moving* objects, *opening/closing* objects, *activating/deactivating* objects, *using* one object on another, and other actions found in other simulated environments [5, 30, 26]. Each action can take zero (e.g. *wait*), one (e.g. *take seed*), or two (e.g. *use spectrometer on plant*) arguments. A list of all actions is shown in APPENDIX B.1. For agents, we include two additional handicap actions to assist with their (generally) poor abilities for navigation: the ability to teleport to specific task-relevant named locations provided in each task (such as the *science lab* or *cafeteria*), and the ability to teleport directly beside any object it is currently (or has previously) observed. Additional details on the teleport action are provided below.

## 3.2 Discovery Task Themes, Difficulty Levels, and Parametric Variations

Specific tasks in DISCOVERYWORLD are parametrically generated from a set of 24 high-level templates. These templates are divided into 8 *discovery task themes* with *3 levels of difficulty* for each theme. A high-level description of the 8 *discovery task themes* is shown in Table 2, with full details (including spoilers) in APPENDIX B.3. For each of the 24 {theme $\times$ difficulty} combinations, the DISCOVERYWORLD simulator is capable of generating a large number of parametric versions of that theme that constitute a particular instance of a *task*. These parametric variations vary the specific task solution, which typically involves dynamically generating new values for specific object properties, and/or substituting in different task objects. For example, in the *Proteomics* theme, each parametric variation of a given task generates a different clustering problem, with different data, that points to a different animal as the solution, and places the animals at different starting locations within the environment. Parametric variation generation is deterministic and controlled by a single *random seed* provided to the task generator. While a large number of specific instantiations of each theme are possible, due to the cost associated with evaluating a large number of tasks, our official benchmark is evaluated on 5 seeds (i.e., random seeds *zero* through *four*) for each theme and difficulty level, resulting in a total of $8 \times 3 \times 5 = 120$ different game instances or *tasks*.

**Task Difficulty:** Difficulty settings *(easy, normal, challenge)* vary both the complexity of the discovery problem, and the complexity of the environment. For example, the *normal* difficulty of *Proteomics* requires successfully clustering two-dimensional data to arrive at a solution, while the *challenge* difficulty requires clustering three-dimensional data. Other tasks similarly increase

---

[3]Text-based games for research typically render observations as lists of objects [5, 30]. Here, we provide the same list-of-object content, but in JSON format.

Table 3: An example scorecard provided by DISCOVERYWORLD for an instance of a *Reactor Lab* task, including *task completion*, *task process*, and *discovered explanatory knowledge* scores.

| Scorecard: Rector Lab, Normal Difficulty, Seed 1 | Out of |
|---|---|
| **Task Completion:** Was the task completed successfully? | **/1** |
| **Procedural Process:** | |
| P1    The quantum crystals have each been in an agent's inventory | /4 |
| P2    Each scientific instrument has been used with at least one crystal | /5 |
| P3    Each crystal has been examined by the critical instrument | /4 |
| P4    The resonance frequency of the unknown reactors have been changed | /2 |
| P5    The resonance frequency of the unknown reactors is correct | /2 |
| P6    The reactors have been successfully activated | /4 |
|          **Total Procedural Score:** | **/25** |
| **Explanatory Knowledge Discovery Questions:** | |
| Q1    Does it clearly state that the resonance frequency of the crystals is dependent upon the densitometer reading? | /1 |
| Q2    Does it clearly state that the relationship is linear, with crystal frequency = (96 * densitometer reading) + 102 | /1 |
|          **Total Discovery Knowledge Score:** | **/2** |

the difficulty of the discovery problem on some critical dimension – for example, the number of substances that have to be mixed, the complexity of the regression problem, or the complexity of rules that govern a pattern of behavior. In terms of environment complexity, both *normal* and *challenge* tasks are generally undertaken on large maps with complex environments, with the *challenge* map typically an extension of the *normal* map. In contrast, the *easy* tasks are generally performed in small, constrained environments – typically small single rooms with few or no distractors, similar to the unit test environments. *Easy* environments also frequently have simplified answering mechanisms, such as using forced-choice tasks (for example, requiring moving a specific kind of object into one of several containers to provide the answer).

**Reduced Spatial Complexity through Teleport:** A common criticism of evaluating complex skills (such as scientific discovery) in embodied virtual environments is that a given agent might be competent at core scientific discovery skills, but poor at skills related to embodiment, such as *picking up* objects, *using* objects, or *spatial tasks* such as navigating from one location to another. If an agent requires these embodiment skills to complete a task, its overall task performance may be an underestimate of its true competency in scientific discovery. We address these challenges in two ways. First, for *spatial and navigation tasks*, we include two types of optional *teleport* actions that can greatly simplify an agent's navigation challenges. The *teleport-to-location* action allows an agent to instantly teleport to a specific location from a list of curated task-relevant locations for each scenario. Similarly, the *teleport-to-object* action allows an agent to instantly teleport to any object that it has previously observed in the environment by name. Finally, for both navigation and other embodiment skills, we provide a set of unit tests (described below) that directly evaluate an agent's competency on skills required to traverse and interact with the environment.

## 3.3   Unit Tests

In addition to the 8 discovery themes, to help disentangle whether a given model's performance is due to a difficulty in completing normal day-to-day tasks within the environment, versus completing tasks requiring scientific discovery skills in particular, we include 10 additional unit test themes that test specific common task competencies. These include a combination of common pick-and-place and navigation tasks (such as those found in ALFWORLD [26], MINIGRID [4], and MINIHACK [23]), as well as DISCOVERYWORLD-themed tasks, such as measuring objects with instruments, or having multi-turn interactions with other agents. The unit test generator is also parametric and capable of generating a large number of specific tasks for each unit test theme. Specific unit test themes are described in detail in APPENDIX B.4, along with example screenshots of the unit test environments.

## 3.4   Evaluation Metrics

To evaluate agents' progress in DISCOVERYWORLD, we devised three automatic metrics: (1) task completion (a binary metric); (2) a fine-grained report card for each task tracking task-relevant actions, to measure partial performance on relevant discovery procedures; (3) the accuracy of discovered explanatory knowledge with respect to a gold reference. Together these allow an agent's progress (including partial progress) to be automatically assessed. An example scorecard is shown in Table 3.

Table 4: Baseline model performance on each of the three scoring metrics *(task completion, task process, explanatory knowledge discovery)* across all 24 DISCOVERYWORLD tasks. Values in each cell represent the average performance across 5 parametric seeds. *Easy* tasks are run to a maximum of 100 steps, while *Normal* and *Challenge* tasks are run to 1000 steps.

| # | Topic | Task | ReACT | | | Plan+Execute | | | Hypothesizer | | |
|---|-------|------|-----------|------------|-----------|-----------|------------|-----------|-----------|------------|-----------|
| | | | Procedure | Completion | Knowledge | Procedure | Completion | Knowledge | Procedure | Completion | Knowledge |
| **Proteomics** | | Clustering | | | | | | | | | |
| 1 | Easy | Simplified Clustering | 0.87 | 0.20 | 0.20 | 0.80 | 0.00 | 0.00 | 0.90 | 0.40 | 1.00 |
| 2 | Normal | Clustering (2D) | 0.88 | 0.40 | 0.40 | 0.68 | 0.20 | 0.00 | 0.93 | 0.40 | 0.40 |
| 3 | Challenge | Clustering (3D) | 0.88 | 0.40 | 0.60 | 0.58 | 0.20 | 0.00 | 0.93 | 0.40 | 0.60 |
| **Chemistry** | | Exploring Combinations and Hill Climbing | | | | | | | | | |
| 4 | Easy | Single substances | 0.87 | 1.00 | 1.00 | 0.70 | 0.60 | 0.40 | 0.90 | 0.00 | 0.40 |
| 5 | Normal | Mix of 3 substances | 0.82 | 0.00 | 0.00 | 0.87 | 0.40 | 0.00 | 0.93 | 0.60 | 0.40 |
| 6 | Challenge | Mix of 4 substances | 0.90 | 0.40 | 0.00 | 0.90 | 0.40 | 0.00 | 0.87 | 0.00 | 0.00 |
| **Archaeology** | | Correlations | | | | | | | | | |
| 7 | Easy | Simple instrument | 0.27 | 0.60 | 0.00 | 0.33 | 0.20 | 0.00 | 0.60 | 0.20 | 0.50 |
| 8 | Normal | Instrument Use | 0.72 | 0.40 | 0.30 | 0.74 | 0.00 | 0.00 | 0.64 | 0.40 | 0.40 |
| 9 | Challenge | Correlation | 0.46 | 0.20 | 0.00 | 0.46 | 0.00 | 0.05 | 0.55 | 0.20 | 0.05 |
| **Reactor Lab** | | Regression | | | | | | | | | |
| 10 | Easy | Slope only | 0.42 | 0.00 | 0.40 | 0.44 | 0.00 | 0.10 | 0.38 | 0.00 | 0.20 |
| 11 | Normal | Linear regression | 0.44 | 0.00 | 0.20 | 0.49 | 0.00 | 0.00 | 0.51 | 0.00 | 0.00 |
| 12 | Challenge | Quadratic regression | 0.43 | 0.00 | 0.20 | 0.39 | 0.00 | 0.00 | 0.39 | 0.00 | 0.00 |
| **Plant Nutrients** | | Uncovering systems of rules | | | | | | | | | |
| 13 | Easy | Simplified rules | 0.80 | 0.20 | 0.20 | 0.70 | 0.20 | 0.20 | 0.60 | 0.00 | 0.00 |
| 14 | Normal | Presence rules | 0.91 | 0.60 | 0.00 | 0.84 | 0.40 | 0.00 | 0.56 | 0.00 | 0.00 |
| 15 | Challenge | Logical Rules | 0.89 | 0.40 | 0.00 | 0.73 | 0.40 | 0.00 | 0.62 | 0.00 | 0.00 |
| **Space Sick** | | Open-ended discovery | | | | | | | | | |
| 16 | Easy | Single instrument | 0.78 | 0.60 | 0.00 | 0.68 | 0.40 | 0.10 | 0.80 | 1.00 | 0.60 |
| 17 | Normal | Multiple instruments | 0.58 | 0.00 | 0.13 | 0.45 | 0.00 | 0.13 | 0.16 | 0.00 | 0.33 |
| 18 | Challenge | Novel instruments | 0.55 | 0.00 | 0.00 | 0.26 | 0.00 | 0.00 | 0.20 | 0.00 | 0.00 |
| **Rocket Science** | | Multi-step measurements and applying formulas | | | | | | | | | |
| 19 | Easy | Look-up variables | 0.33 | 0.00 | 0.00 | 0.53 | 0.00 | 0.07 | 0.13 | 0.40 | 0.00 |
| 20 | Normal | Measure 2 variables | 0.51 | 0.00 | 0.05 | 0.34 | 0.00 | 0.00 | 0.11 | 0.00 | 0.00 |
| 21 | Challenge | Measure 5 variables | 0.43 | 0.00 | 0.00 | 0.15 | 0.00 | 0.00 | 0.22 | 0.00 | 0.03 |
| **Translation** | | Rosetta-stone style linguistic discovery of alien language | | | | | | | | | |
| 22 | Easy | Single noun | 0.40 | 0.40 | 0.20 | 0.30 | 0.00 | 0.00 | 0.20 | 0.20 | 0.00 |
| 23 | Normal | Noun and verb | 0.20 | 0.00 | 0.00 | 0.68 | 0.40 | 0.00 | 0.84 | 0.40 | 0.00 |
| 24 | Challenge | Noun, adj., and verb | 0.49 | 0.00 | 0.00 | 0.55 | 0.20 | 0.05 | 0.15 | 0.00 | 0.00 |
| **Average (Easy)** | | | 0.59 | 0.38 | 0.25 | 0.56 | 0.18 | 0.11 | 0.56 | 0.28 | 0.34 |
| **Average (Normal)** | | | 0.63 | 0.18 | 0.14 | 0.64 | 0.18 | 0.02 | 0.58 | 0.23 | 0.19 |
| **Average (Challenge)** | | | 0.63 | 0.18 | 0.10 | 0.50 | 0.15 | 0.01 | 0.49 | 0.08 | 0.08 |

Two of these three metrics (*task completion*, and *task process*) are measured automatically by DISCOVERYWORLD. For the third, *discovered explanatory knowledge*, the scorecard provides specific binary questions to answer with reference to knowledge that an agent has produced (in its explanations, memories, reports, or other knowledge structures). These can either be graded manually by a human, or provided to a language model to grade. DISCOVERYWORLD provides code for automatic grading using OPENAI models, and in our evaluation we make use of GPT-4O, a long-context (128k token) model that allows even large knowledge structures to be graded. Examples of both positive and negative knowledge assessments are provided in APPENDIX D.

# 4 Experiments, Baseline Agents, and Human Baselines

In this section we examine the performance of strong baseline agents on each of the DISCOVERY-WORLD tasks. In addition, we investigate the performance of human scientists, and highlight the performance gap between current agent models and real human scientists.

## 4.1 Experimental setup

For the purposes of this work, we seek to better understand the zero-shot generalization performance of AI agents on tasks that require iterative scientific discovery: that is, coming up with hypotheses, doing in-game experiments to (in)validate them, then arriving at a discovered solution for the task. As such, we evaluate the performance of three contemporary baselines in a zero-shot setting (though *single-task, multi-task, or curriculum learning* settings with separate training and evaluation sets are

Table 5: Baseline model performance on two relevant scoring metrics *(task completion, task process)* across all 10 unit test tasks. Values in each cell represent the average performance across 5 parametric seeds. Unit tests tasks are run to a maximum of 100 steps. As the unit tests evaluate basic skills instead of discovery skills, explanatory knowledge is not evaluated.

| # | Unit Test Topic | ReACT Procedure | ReACT Completion | Plan+Execute Procedure | Plan+Execute Completion | Hypothesizer Procedure | Hypothesizer Completion |
|---|---|---|---|---|---|---|---|
| 25 | Multi-turn dialog with an agent | 1.00 | 1.00 | 1.00 | 1.00 | 1.00 | 1.00 |
| 26 | Measure an object with an instrument | 0.87 | 0.60 | 0.73 | 0.40 | 1.00 | 1.00 |
| 27 | Pick-and-place object | 0.90 | 0.80 | 0.80 | 0.60 | 1.00 | 1.00 |
| 28 | Pick-and-give object | 1.00 | 1.00 | 1.00 | 1.00 | 1.00 | 1.00 |
| 29 | Read DiscoveryFeed posts | 1.00 | 1.00 | 0.90 | 0.80 | 1.00 | 1.00 |
| 30 | Move through doors | 0.58 | 0.20 | 0.25 | 0.00 | 0.30 | 0.00 |
| 31 | Using keys with doors | 0.69 | 0.20 | 0.54 | 0.00 | 0.69 | 0.00 |
| 32 | Navigate to a specific room in a house | 0.20 | 0.20 | 0.20 | 0.00 | 0.20 | 0.20 |
| 33 | Search an environment for an object | 0.80 | 0.80 | 0.60 | 0.60 | 1.00 | 1.00 |
| 34 | Interact with a moving agent | 0.60 | 0.20 | 0.53 | 0.00 | 0.53 | 0.20 |
| **Average (Unit Tests)** | | 0.76 | 0.60 | 0.66 | 0.44 | 0.77 | 0.64 |

possible with this benchmark; see APPENDIX B.2 for these configurations). In the zero-shot setting, an agent has no prior exposure to DISCOVERYWORLD, and is evaluated on all 120 tasks. Each task is evaluated independently, without carry-over knowledge from one task to another.

## 4.2 Baseline Agent Models

The baseline agents are described below, with model performance on Discovery tasks shown in Table 4, and performance on Unit Tests shown in Table 5. We use the GPT-4O model for all our agents due to its higher performance and lower cost compared to other models available. For space we provide high-level descriptions of each model here, with additional implementation details and run costs provided in APPENDIX C.

**ReAct**: This agent uses the ReAct [35] approach of generating a thought and action at each step given the recent trajectory of thoughts, actions and observations. Each action is executed in the environment and the observation is added to the trajectory. In addition to this trajectory, we also provide the current game state information as text, e.g., nearby objects, teleportable locations, etc. If needed, we trim the trajectory (remove oldest steps first) to fit the prompt within the maximum token limit, which (in practice) included up to the last 40 steps of the trajectory. To evaluate this agent's discovered knowledge, we evaluate the concatenation of the agent's "thoughts" across all time steps.

**Plan+Exec**: Since the ReAct trajectories can be very long and lead to errors due to distractors, we also evaluate a plan-and-execute [29, 34] approach. We use the LLM to generate a plan, and each step of the plan is independently executed using the same ReAct agent as above. Since each plan step is simpler than the task, their ReAct trajectories are much smaller, reducing the distractors. Discovery tasks require an iterative planning approach to adapt to new findings, so we use iterative decomposition [11] to only generate one step of the plan based on the previous planning steps and their success. Discovered knowledge is evaluated as with the ReAct agent from the execution steps.

**Hypothesizer**: This agent resembles the architecture of CLIN [18], with the agent keeping an explicit working memory of running hypotheses and measurements that are updated after taking each action, and conditioning its next actions on the contents of this memory. This explicit knowledge store allows more directly evaluating an agent's discovered knowledge (with an example of Hypothesizer's knowledge provided in subsection C.3). The agent similarly maintains a brief plan and running hypothesis, and is prompted to explain how its action progresses the plan to evaluate that hypothesis.

**Text only vs. Multi-Modal:** For this baseline evaluation, both REACT and PLAN+EXEC models are *text-only*, in that they include only text (or JSON-formatted) input. The HYPOTHESIZER agent includes both text observations as well as visual observations (i.e. screen frames, as provided by the API) as input.

Table 6: Expert human scientist performance on DISCOVERYWORLD tasks, as well as average task completion time. Scores represent average performance of up to 11 humans when playing the same seed of a discovery task.

| | | | Human Performance | | | | | | | |
|---|---|---|---|---|---|---|---|---|---|---|
| # | Topic | Difficulty | Procedural | Completion | Knowledge | Avg. Steps | Movement Steps | Action Steps | Prop. Action Steps | # Samples |
| 2 | Proteomics | Normal | 0.90 | 0.80 | 0.90 | 277 | 262 | 15 | 0.06 | 10 |
| 3 | Proteomics | Challenge | 1.00 | 1.00 | 1.00 | 203 | 192 | 11 | 0.05 | 10 |
| 5 | Chemistry | Normal | 0.98 | 0.90 | 0.64 | 369 | 293 | 76 | 0.23 | 10 |
| 6 | Chemistry | Challenge | 0.95 | 0.89 | 0.77 | 401 | 324 | 76 | 0.21 | 9 |
| 8 | Archaeology | Normal | 0.92 | 1.00 | 0.91 | 310 | 275 | 35 | 0.14 | 10 |
| 9 | Archaeology | Challenge | 0.77 | 0.36 | 0.09 | 276 | 240 | 36 | 0.13 | 11 |
| 11 | Reactor Lab | Normal | 0.78 | 0.60 | 0.36 | 414 | 340 | 74 | 0.18 | 10 |
| 12 | Reactor Lab | Challenge | 0.70 | 0.33 | 0.25 | 281 | 236 | 45 | 0.16 | 9 |
| 14 | Plant Nutrients | Normal | 0.93 | 1.00 | 0.64 | 365 | 310 | 55 | 0.15 | 10 |
| 15 | Plant Nutrients | Challenge | 0.88 | 0.70 | 0.32 | 358 | 306 | 52 | 0.16 | 10 |
| 17 | Space Sick | Normal | 0.69 | 0.73 | 0.59 | 2111 | 1958 | 153 | 0.08 | 11 |
| 18 | Space Sick | Challenge | 0.60 | 0.11 | 0.11 | 3458 | 2988 | 470 | 0.13 | 9 |
| 20 | Rocket Science | Normal | 0.58 | 0.30 | 0.40 | 274 | 240 | 34 | 0.13 | 10 |
| 21 | Rocket Science | Challenge | 0.57 | 0.11 | 0.33 | 487 | 334 | 153 | 0.36 | 9 |
| 23 | Translation | Normal | 0.79 | 0.73 | 0.77 | 1033 | 948 | 86 | 0.07 | 11 |
| 24 | Translation | Challenge | 0.62 | 1.00 | 0.68 | 859 | 794 | 65 | 0.07 | 11 |
| **Average (Human)** | | **Normal** | 0.82 | 0.76 | 0.65 | 644 | 578 | 66 | 0.13 | 10 |
| **Average (Human)** | | **Challenge** | 0.76 | 0.56 | 0.44 | 790 | 677 | 113 | 0.16 | 10 |

## 4.3 Human Evaluation

To compare model performance against human performance, we recruited 11 practicing human scientists to complete the DISCOVERYWORLD tasks, with their performance shown in Table 6. Scientists were recruited on the UPWORK platform, each with: (1) an MSC or PHD in a natural science, (2) self-evaluated comfort and fluency with statistical methods and common software like spreadsheets, (3) comfort and previous experience with 2D top-down games. Additional details regarding human participant experiments are provided in APPENDIX E.

## 5 Discussion

**Human Performance**  As shown in Table 6, discovery task completion rates varied from tasks that were solved by all participants, to several *Challenge* difficulty tasks that were solved by only a single participant, highlighting the range of difficulties found across task themes, and range of expertise provided by each scientist. Overall, the average completion rate across tasks was 66%, with 11 of 16 tasks performed by humans having completion rates exceeding 60%. Average knowledge performance was slightly lower at 55%, reflecting that when stuck, the humans sometimes attempted brute force solutions to tasks that may have eventually yielded correct answers (e.g. trying all possible crystal frequencies, if they didn't use regression to fit the data), but without producing the required explanatory discovery.

**Agent Performance**  In contrast to human scientists, the baseline agent scientists exhibited poor overall performance, as shown in Table 4. The most performant discovery agent (REACT) completed 38% of *easy* tasks and 18% of *challenge* tasks, while the agent that best discovered explanatory knowledge (HYPOTHESIZER) discovered only 34% of gold knowledge in *easy tasks*, and only 8% in *challenge* tasks. An analysis of these agents performance on the Unit Test tasks (Table 5) shows moderate overall performance, with completion rates in the 60%+ range, suggesting that current agents are competent at many of the components of scientific discovery – like measuring objects with instruments – but currently lack the capacity to perform end-to-end discovery in most DISCOVERYWORLD settings.

# 6 Conclusion

We have presented DISCOVERYWORLD, the first virtual environment for developing and benchmarking an agent's ability to perform end-to-end scientific discovery. Each of the 120 tasks requires an agent to form hypotheses, design and run experiments, analyze results, and act on conclusions. We empirically demonstrate that expert human scientists find the challenge tasks in DISCOVERYWORLD difficult but solvable, while strong agent baselines struggle to complete most tasks, or discover critical explanatory knowledge. We hope DISCOVERYWORLD will inspire and accelerate the development of new, general AI discovery agents by the community.

# 7 Limitations and Broader Impacts

**Simulation Fidelity:** DISCOVERYWORLD is inherently a low-fidelity representation of the physical world, with an abstracted action space. As such, agents that perform well on DISCOVERYWORLD may not necessarily perform well at making discoveries in the real world, given the much larger search and action space. That being said, while the simulated environment is low-fidelity compared to the physical world, the steps of discovery that are simulated (from ideation, hypothesis formation, experiment design and execution, data analysis, and forming conclusions) are common steps in the scientific method regardless of whether those actions are taking place in the real world or a virtual environment.

**Agent Cost:** The cost of LLM inference is rapidly decreasing, as evidenced by OPENAI recently releasing a multi-modal model (O1-MINI) that costs approximately $0.15/1M$ input tokens, a decrease of almost 20 times over the GPT-4O base model used in this work (current cost: $2.50/1M$ input tokens), which was the most performant model at the time of submission. At submission time, these GPT-4O agent models were quite costly, ranging from approximately USD$3k-$10k to run for the complete set of 120 tasks in DISCOVERYWORLD, though their cost has since decreased. This cost is due in large part to (1) the long (1000+ steps) runtimes, with each step requiring at least one LLM call, (2) the large number of individual tasks to evaluate, and (3) the use of performant but costly API-BASED models that charge per token. We believe that developing inexpensive models that allow for rapid iteration is a clear near-term goal to help facilitate developing discovery agents that must perform long-horizon tasks in this 1000+ step range. A recommended **reduced-cost evaluation** is to evaluate on only the *easy* difficulty tasks, while limiting runtime to *100 steps* per run. We estimate (using the cost estimates in Appendix C.4) such a setup would cost approximately $50-100 for the REACT agent at current GPT-4O pricing, and potentially significantly less with other base models.

**Societal Benefits and Risks:** Automated scientific discovery has the potential for broadly positive societal impact by accelerating the pace of scientific progress, potentially decreasing the time it takes for novel discoveries in medicine, chemistry, technology, and other areas of broad impact. If discovery systems are used by individuals or groups with prosocial intentions, there is the potential for broad societal impact. Conversely, if individuals or groups with negative intentions choose to attempt to accelerate discoveries that may be harmful, there is the potential for those individuals or groups to cause harm using automated scientific discovery models.

# Acknowledgments and Disclosure of Funding

We thank members of the Aristo and Semantic Scholar teams at the Allen Institute for Artificial Intelligence for their helpful comments and support throughout the development of this work. Game assets (e.g., sprites) used in DISCOVERYWORLD are in large part from the CUTERPG pack by Pixy-Moon that was purchased for this work and requires attribution, with some additions or modifications from the authors and OPENAI DALL-E for science-themed content.

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

# A  Overview

The supplementary materials is structured as follows:

- **Appendix B: Additional Details on DISCOVERYWORLD**. This section covers the *Action Space* and common *Object Properties* of DISCOVERYWORLD, the recommended practices for different *Experimental Settings*, more details on the *Discovery Themes* and the *Unit Tests*.
- **Appendix C: Additional Baseline Model Details**.
- **Appendix D: Automatic Evaluation of Explanatory Discovery Knowledge**.
- **Appendix E: Human Scientist Participants**.

# B Additional Details on DISCOVERYWORLD

To encourage the community to work on general AI discovery agents, we have open-sourced DIS-COVERYWORLD under Apache-2.0 license on GitHub at github.com/allenai/discoveryworld.

## B.1 Action Space

The action space for DISCOVERYWORLD is shown in Table 7.

Table 7: The action space for DiscoveryWorld, which includes 14 main actions. Actions can take zero (e.g. WAIT), one (e.g. TAKE THERMOMETER), or two (e.g. USE SPECTROMETER ON SOIL) arguments. Two handicap actions (∗) are available to agents to assist with poor spatial/navigation abilities.

| Action | Description | Action | Description |
|--------|-------------|--------|-------------|
| MOVE DIR | Move North | USE OBJ [ON OBJ] | Use spectrometer on soil |
| TAKE OBJ | Take thermometer | EAT OBJ | Eat mushroom |
| DROP OBJ | Drop seed | READ OBJ | Read rocketry book |
| PUT/GIVE OBJ TO OBJ | Put sample in jar | WAIT | Do nothing |
| OPEN/CLOSE OBJ | Open door | FEED | View DiscoveryFeed |
| DE/ACTIVATE OBJ | Activate pendulum | TELEPORT LOC ∗ | Teleport to Science Lab |
| TALK OBJ | Talk to Colonist | TELEPORT OBJ ∗ | Teleport to Microscope |

## B.2 Experimental Settings

As both a development environment and benchmarking tool, we provide the following recommended configurations for evaluating on DISCOVERYWORLD in common experimental settings:

**Zero-shot:** For zero-shot evaluation, an agent should not have prior exposure to DISCOVERY-WORLD, and can be evaluated on all 120 tasks. Each task should be evaluated *independently*, without any carry-over knowledge from one task to another. *All of the baseline agent models provided in this work are evaluated zero-shot.*

**Single-Task Learning:** For agents that require training data (e.g., RL agents, few-shot examples, etc.) of highly similar tasks (i.e. within-task designs), following the TEXTWORLD [5], ALF-WORLD [26], and SCIENCEWORLD [30] conventions, we recommend the following *within-theme* design: training on parametric seeds 0 and 1, developing on seed 2, and testing on seeds 3 and 4. In addition, all tasks in the *Unit Tests* are available for training.

**Multi-Task Learning:** For agents that require training data (e.g., RL agents, few-shot examples, etc.), following the JERICHO benchmark [7] for evaluating text game suites, we recommend training and evaluating *across themes*. Tasks 1 to 6 (i.e., theme *Proteomics* and *Chemistry*) can be used for training, tasks 7 through 12 (*Archaeology* and *Reactor Lab*) for development, and the remaining tasks 13 through 24 (*Plant Nutrients, Space Sick, Rocket Science*, and *Translation*) for testing. In addition, all tasks in the *Unit Tests* are available for training.

**Curriculum Learning:** For agents that require training data of growing complexity, irrespective of the experimental setting, we recommend to simply follow the natural progression of the difficulty levels for each theme. In addition, all tasks in the *Unit Tests* are available for training.

## B.3 Discovery Themes

The following are descriptions of the 8 main discovery themes, and how these themes vary across each of the 3 levels of difficulty. *Note: This section contains spoilers!*

**Proteomics** This theme examines an agent's ability to perform analyses that involve clustering and identifying outliers. The agent is told that a number of animal species exist in a remote habitat, with one hypothesized to have migrated from an isolated island in the distant past – with the agent required to measure the abundance of different proteins (using a *protein meter*) to determine which

species is the outlier. In *normal* difficulty the clustering problem is two dimensional (that is, values of two proteins are provided for each measured animal), while in *challenge* difficulty, the clustering problem is three dimensional. Protein concentrations are provided as values between zero and one. The protein concentration data is manufactured such that all animals protein values exist on the surface of a sphere centered on a randomly chosen point, with inliers having a small radius from the center (e.g. $0.1$), and outliers having a much farther radius (e.g. $0.4$). Inliers and outliers are clearly visible on a plot by inspection. The *easy* difficulty level reduces the number of animals from 5 to 3, and has the entire task take place in a small lab with unmoving animals, rather than a large map with moving animals.

**Chemistry**    The chemistry theme evaluates an agents ability to explore a large solution space in the presense of a hill-climbing signal. Here, the agent is told they must manufacture a chemical (a rust removal agent), from some combination of unique chemicals available to them. The agent must enumerate the possible combinations of chemicals, prepare these solutions, and test them on a specific rusted object. After being immersed in a solution, the rusted object will decrease from *heavily rusted* to *moderately* or *lightly rusted* based on the cosine similarity between the chemical concentration of the solution that it is immersed in, and the correct solution, providing a hill-climbing signal for agents to use to more quickly explore the space of possible solutions. The *normal* difficulty has three possible chemical dispensers, the *challenge* has four chemical dispensers, and both require agents to find specific concentrations of chemicals (e.g. *1 part Chemical A, 2 parts Chemical B*). The *easy* difficulty has four chemical dispensers, but requires no mixing – one of the four chemicals is a direct solution.

**Archaeology**    The archaeology theme involves measuring (and validating) artifacts using radioisotope dating. In the *challenge* difficulty, the agent is presented with a dig site that contains six artifacts – three known, three unknown – as well as a radioisotope meter that measures 4 different radioisotopes. The agent is prompted that it is unknown if radioisotope dating works on *Planet X*, and if it does, which of the 4 radioisotopes will be useful for dating. The agent is tasked with finding the oldest unknown artifact, and placing a red flag beside its dig site. The known artifacts include one from the stone age (a stone hammer), one from the bronze age (a bronze chisel), and one from the iron age (iron tongs). The agent must know that in radioisotope dating, older artifacts have lower concentrations of a given radioisotope due to radioactive decay. The agent must also make the critical insight that stone age artifacts are older than bronze age artifacts, and iron age artifacts are younger than bronze age artifacts. The data are manufactured such that one radioisotope follows this pattern, while others don't (with a correlation of $R^2 < 0.1$ between age and radioisotope value for all incorrect channels). In the *normal* difficulty, the instrument directly supplies age, and evaluates instrument use rather than instrument validation. The *easy* difficulty is similar to *normal* but in a small lab environment.

**Reactor Lab**    The reactor lab theme requires agents to discover and use mathematical relationships between measured quantities using regression. The reactor lab contains a number of *quantum crystals*, each with a specific resonance frequency. The agent must place all crystals into specific slots, and tune each slot to that crystal's frequency, for the reactor to activate. The frequency depends upon one of five measured properties – density, temperature, radioactivity, crystal size, or crystal spectra – which can be measured using instruments in the adjoining science lab. The agent is provided with some number of crystals with known frequencies, and two crystals with unknown frequencies. They must measure each crystal with each of the instruments, determine which physical property is related to the frequency (as well as how), and use this equation to calculate the frequencies of the unknown crystals. In the *normal* setting, 2 known crystals are provided, and the relationship can be found with linear regression ($y = mx + b$). In the *challenge* setting, 3 known crystals are provided, and the relationship is quadratic ($y = a^2x + bx + c$). In the *easy* setting, only a single (correct) instrument is provided, there is only one unknown crystal, and the relationship requires only inferring the slope ($y = mx$).

**Plant Nutrients**    The plant nutrient theme requires agents to infer systems of rules based on observing both positive and negative examples. The agent finds itself at a botanical research center, and given the task of identifying the (unusual) nutrients that plants on PLANET X prefer. The research center includes a "pilot field" where 12 seeds have been planted, some of which sprouted correctly, as well as 3 "test fields", each controlled by a soil computer that allows manipulating the nutrients in its soil. The agent must infer the necessary levels of nutrients in the soil from the pilot field, then

successfully grow at least two new plants by setting the nutrient levels in a test field to be appropriate, and planting (and growing) seeds in that field. All settings include five different nutrients. In the *normal* setting, rules are simple presence-at-value rules (i.e. plants require a specific nutrient, at a specific value of either *low, medium,* or *high* to grow), and these can be inferred from positive examples alone. In the *challenge* setting, the rules involve logical relations (e.g. XOR, AND, OR, NOT) between nutrients, and more examples are provided. The *easy* setting resembles *normal* except that nutrients are binary (*present/absent*) rather than varying in concentration.

**Space Sick**    The space sick theme requires agents to use open-ended discovery skills to investigate the cause of an illness. The agents are tasked with identifying why some colonists are becoming mildly ill when eating local food, and correcting this. The agent must discover that some food has been contaminated by mold, which can be directly observed with some instruments, and indirectly observed with others (e.g. through elevated spectrometer readings), or indirectly inferred (e.g. cooking food causes it to become safe). Distractors (such as some food being slightly radioactive) lead the agent down paths that do not solve the task. In the *normal* difficulty, the mold is directly observable using instrumentation. In the *challenge* difficulty, the agent must discover novel detection instrumentation. In the *easy* setting, the agent is placed in a lab with 3 samples of food and 4 instruments, and must identify which food sample is contaminated (which is detectable by only a single instrument).

**It's (not) Rocket Science!**    The rocket science theme requires agents to measure quantities in an environment, then substitute these quantities into known equations to complete a task. Here, the agent is tasked with sending a rocket into a specific orbital height around PLANET X. In the *challenge* version of this task, the agent needs to enter to appropriate orbital velocity, as well as the type (and quantity) of propellant to be used. The agent is provided with a rocketry book containing all the formulas needed to complete the task. However, before it can calculate the required quantities, it needs to infer out how to measure unknown values such as PLANET X's gravity and radius, as well as properties related to the three available types of propellant such as density, mass flow rate, and thrust when consumed by the rocket engine. This theme involves recalling known experiments performed on Earth and transposing them to PLANET X (e.g., Eratosthenes' classical experiment to measure Earth's radius[4]). The *normal* difficulty simplifies the task to requiring only orbital velocity (not propellant), while the *easy* difficulty tells the agent they are on a planet with a similar mass and radius to a known celestial body (e.g. Earth, Mars, Venus), eliminating the requirement to measure these values of PLANET X.

**Lost in Translation**    The translation theme requires agents to infer the meanings of progressively more complicated utterances given access to an environment containing Rosetta-stone-style information. Here, the agent must translate an unknown language used by the native inhabitants of PLANET X. The agent first talks to one of the inhabitants, who speaks an utterance. From this, the agent must explore the village to gather clues about what each word means (e.g., visiting shops will help identifying the name of the items being sold there). In the *challenge* difficulty, instructions are composed of a verb, an amount, a color, and an item name (e.g., bring 3 red flowers). Bringing the correct number of the correct objects back to the inhabitant will complete the task. In the *normal* difficulty, the utterance requires translating only a single specific object. In the *easy* setting, the task is reduced to a small lab setting, where the agent must identify which item to bring to another agent based on its translation (accessible by a sign near the object).

### B.4    Unit Tests

### B.4.1    Unit Test Enviroments

The unit tests are shown in Figure 3, and described in Table 8.

### B.5    Object Properties

A list of common object properties provided by the base OBJECT storage class is shown in Table 9. Note that specific objects (like a QUANTUMCRYSTAL) may implement properties not included in the base class (such as *resonanceFrequency*).

---

[4]https://en.wikipedia.org/wiki/Eratosthenes#Measurement_of_Earth's_circumference

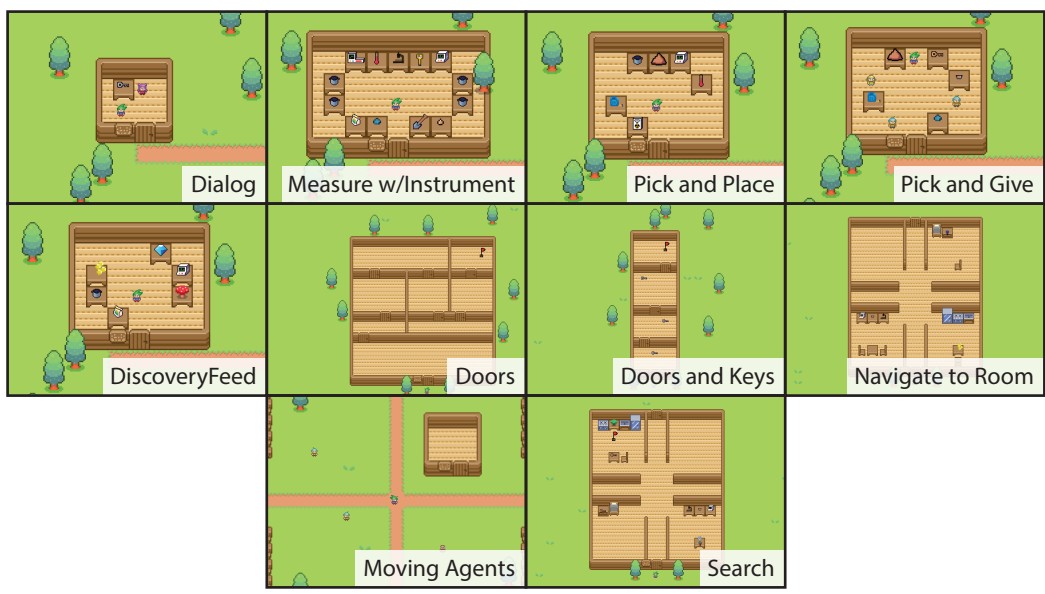

Figure 3: Example instances of the 10 Unit Test themes.

Table 8: High-level descriptions of the 10 *unit test themes* in DISCOVERYWORLD.

Specific unit tests tasks are parametrically generated from a given theme.

| # | Theme | Description |
|---|-------|-------------|
| 25 | Multi-turn dialog with agent | The user agent must talk to an NPC agent, and respond correctly to its requests. The NPC agent asks the user agent to select the dialog item that says (for example) 'kiwi', and the user agent must select this dialog option in the dialog tree amongst 8 distractors. The task completes when this process happens successfully 3 times. |
| 26 | Measure an object with an instrument | The user agent must take a specific item (from 4 possible items), measure a specific property (from 5 possible instruments), and place the item in one of 4 containers depending on the range (e.g. temperature range 0-2.5C in Container A, temperature range 2.5-5.0C in Container B, etc.). |
| 27 | Pick-and-place object | The user agent must take a specific item (from 5 possible items) and place it in a specific container. |
| 28 | Pick-and-give object | The user must take a specific item (from 5 possible items) and give it to a specific NPC agent (from 3 possible agents). |
| 29 | Read DiscoveryFeed posts | The user agent must take a specific item (from 5 possible items), and place it in a specific container. The item to take is provided in a DiscoveryFeed post. |
| 30 | Move through doors | The user agent must successfully navigate a randomly generated maze (in the form of a house with walls/doors in random locations), and pick up a flag at the end of the maze. All doors must be opened and the flag must be taken for task success. |
| 31 | Using keys with doors | The user agent must successfully navigate a randomly generated house that contains 3 doors, and pick up a flag in the final room. The doors require keys, which are randomly placed near the doors. |
| 32 | Navigate to a specific room in a house | The user agent must succesfully navigate to a specific room in a house (such as the *kitchen* or *bedroom*), which it must infer based on the contents of each room. Once in the correct room, it signifies its selection by dropping a flag. |
| 33 | Search an environment for an object | The user agent must navigate the interior and exterior of a house environment in search of a red flag. |
| 34 | Interact with a moving agent | The user agent must interact with 3 moving NPC agents spread throughout the environment, notifying each that it's about to rain, and they may want to return home to avoid becoming wet. |

Table 9: A list of object properties common to the OBJECT storage class in DISCOVERYWORLD. From this list is omitted any properties tied to a specific object, e.g., the fuel quantity in the rocket.

| # | Property Name | Description |
|---|---|---|
| 1 | grid location | Initial world location (X, Y) |
| 2 | isMovable | Can it be moved? |
| 3 | isPassable | Can an agent walk over this? |
| 4 | obscuresObjectsBelow | Does it obscure/hide objects on layers below it? |
| 5 | isActivatable | Is this a device? (more specifically, can it be activated/deactivated?) |
| 6 | isActivated | Is this device currently activated? |
| 7 | isUsable | Can this device be used with another object? (e.g., specifically through the 'use' action). |
| 8 | isDialogable | Can it be dialoged with? |
| 9 | isShovelable | Can it be shoveled? |
| 10 | isReadable | Can it be read? |
| 11 | document | Any text to read |
| 12 | temperatureC | The default object temperature, in Celsius |
| 13 | heatSourceMaxTemp | If it is a heat source, then this is the maximum temperature that it can reach |
| 14 | coolSourceMinTemp | If it is a cool source, then this is the minimum temperature that it can reach |
| 15 | isLiving | Is it alive? |
| 16 | substanceName | Name of the substance |
| 17 | isSubstance | Is it a substance? |
| 18 | isAutoReacting | Does it react automatically with other substances? |
| 19 | mixtureDict | Dictionary of substances and their proportions in the mixture |
| 20 | requiresKey | If it requires a key to open/use, then this is a special ID for the key. If the value is <=0, then it doesn't require a key. |
| 21 | keyID | If this object acts as a key, here's its ID (0 by default) |
| 22 | radiocarbonAge | Radiocarbon dating age (in years). -1 means it's not applicable/inconclusive. |
| 23 | radioisotopeValues | Radioisotope values. If empty, then it's not applicable/inconclusive. |
| 24 | soilNutrients | Soil nutrients. If empty, then it's not applicable/inconclusive. |
| 25 | needsNutrientLevels | For seeds/plants: What nutrient levels do they need to grow? |
| 26 | antirequirementsNutrientLevels | A list of dictionaries, each containing a list of nutrient levels under which the seed/plant will NOT grow |
| 27 | density | Object density (in g/cm³). <=0 means it's not applicable/inconclusive. |
| 28 | microscopeModifierText | Modifier text, if the object is viewed under a microscope. This is a list of strings, which are displayed in the microscope view. |
| 29 | microscopeDesc | Short description to be displayed under a microscope. |
| 30 | color | Color description of the object. |
| 31 | spectrum | Spectrum data of the object. |
| 32 | ph | pH value of the object. |
| 33 | radiationusvh | Radiation level in microsieverts per hour. |
| 34 | nitrogen | Nitrogen content. |
| 35 | phosphorus | Phosphorus content. |
| 36 | potassium | Potassium content. |
| 37 | cosCanBeLiquid | Can it exist in liquid form? |
| 38 | cosCanBeSolid | Can it exist in solid form? |
| 39 | cosCanBeGas | Can it exist in gas form? |
| 40 | cosMeltingPointC | Melting point in Celsius. |
| 41 | cosBoilingPointC | Boiling point in Celsius. |
| 42 | cosCombustionPointC | Combustion point in Celsius. |
| 43 | livingMinTemp | Minimum temperature for living organisms. |
| 44 | livingMaxTemp | Maximum temperature for living organisms. |
| 45 | isContainer | Is it a container? |
| 46 | isOpenable | If it's a container, can you open/close it? |
| 47 | isOpenContainer | If it's a container, then is it open? |
| 48 | containerPrefix | Container prefix (e.g., "in" or "on"). |
| 49 | isOpen | Closed by default. |
| 50 | contentsVisible2D | If it is a container, do we render the contents in the 2D representation, or is that already handled (e.g., for pots/jars, that render generic contents if they contain any objects). |
| 51 | contents | Contents of the container (other objects). |
| 52 | parentContainer | Back-reference for the container that this object is in. |
| 53 | parts | List of parts that this object is made of. |
| 54 | subparts | Subparts of the object. |
| 55 | isEdible | Can it be eaten? |
| 56 | isCooked | Is it cooked? |
| 57 | isPoisonous | Is it poisonous? |
| 58 | isPassage | Is this a passage? |
| 59 | materials | List of materials that this object is made of. |
| 60 | manualMaterialNames | A list of material types to add during initialization (in code, rather than from the spreadsheet). |
| 61 | actionHistory | Agent action history, if applicable (None by default). |
| 62 | isAgent | Is this object an agent? (e.g., a person). |
| 63 | isNPC | Is this agent an NPC? |

## C  Additional Baseline Model Details

All models and their implementations are provided in the DISCOVERYWORLD code repository.

### C.1  ReAct

The ReAct agent generates a thought and action at each step given a prompt describing the environment and the previous steps. The prompt contains information about the environment, how to interact with the environment, current environment state, teleportable locations, interactable objects, and valid actions. The previous steps are shown as a sequence of (thought, action, observation)s, e.g.:

```
History of action-observations:
  Action:
    ```json
    {
    "action": "ROTATE_DIRECTION",
    "arg1": "north",
    "thought": "I need to face north to approach the door and enter the building."
    }
    ```
  Observation:
    ```json
    {"message": "I rotated to face north.", "errors": [], "success": true}
    ```
  Action:
    ```json
    {
    "action": "OPEN",
    "arg1": 2056,
    "thought": "I need to open the door in front of me to enter the building."
    }
    ```
  Observation:
    ```json
    {"message": "I opened the door.", "errors": [], "success": true}
    ```
```

The agent generates thoughts and actions using the JSON formatted output as shown above. Each action is executed in the environment and the returned observation is then added to the trajectory. Since tasks in DiscoveryWorld can be extremely long, the trajectories can often exceed the maximum token limit of our LLM. We generally hit the 8K token limit of GPT4-O in 40 steps. To avoid this, we limit the trajectory in the prompt to a maximum of 10K characters by removing the oldest (thought, action, observation)s and replacing it with the string [TRIMMED HISTORY].

### C.2  Plan-and-Execute

Another approach to deal with long trajectories used in prior work [34, 20] is to decompose the problem into a plan of simpler steps and executing each step independently using ReAct. Since it would be challenging to generate the entire plan in one shot *without exploring the environment*, we use iterative decomposition [11] where we generate one step of the plan, then execute it and based on the result generate the next step. The planner uses the same base LLM with the prompt describing the format of the plan, e.g.,

```
Task: Open the door and exit the building.

Plan:
Step 1: Go to the door that leads to the exit
 -- Task completed! I am at the door that leads to the exit
```

```
Step 2: Open the door
 -- Task failed! Door is locked and I don't have the key
Step 3: Find the key to the door
...
```

---

To execute each step of the plan, we use the same ReAct agent as above but additionally add the previous steps of the plans and their results to the prompt. To prevent the ReAct agent from wasting time on infeasible steps, we add a hint to the prompt to consider returning the `Task failed!` message (e.g. Step 2 above) when $1/5^{th}$ of the environment step budget is used up. We use the same truncation strategy for the ReAct agent and no truncation is needed for the planner due to the much shorter plans.

### C.3 Hypothesizer

This agent maintains a working memory of science-related knowledge (allowing the knowledge this agent has discovered to be more directly evaluated). To assist in planning and execution, this agent also maintains a running hypothesis for the task solution, as well as a short natural language plan for the immediate steps it should take. More specifically, at each step, HYPOTHESIZER chooses an action based on it's current working memory, plan, and running hypothesis. It then reflects on the results of that action, and updates the working memory based on the results. The working memory is science themed, storing two types of records: (1) HYPOTHESES, which include the *hypothesis statement*, it's current *status* (i.e. whether it has been confirmed, rejected, or is still pending), and a list of *supporting evidence* the agent has listed as helping make this determination, (2) and MEASUREMENTS, which take the form of specific observations the agent has made of objects in the environment – for example, that a particular plant appeared to grow in a mixture including a specific nutrient. This memory and reflection is similar to CLIN [18], except: (a) this memory is structured to hold science-domain relations, where CLIN's holds causal relations, and (b) this memory is updated after each action in the environment, where CLIN's is updated only after running an entire task to completion. This means that, if an agent takes $N$ steps in an environment, it requires $2N$ LLM calls to accommodate this *act-then-reflect* cycle. Because the memory can grow large, after every 10 actions, the working memory is summarized through a separate LLM call that requests a maximum of 40 separate *hypothesis* or *measurement* entries, and a maximum of $2k$ tokens for the entire memory. When producing actions, HYPOTHESIZER is prompted to produce (1) a specific action, (2) an explanation that includes its running hypothesis, and how this action helps move it towards evaluating this hypothesis, and (3) a short memory describing its plans or any other helpful information it wishes to pass on to the next step. When prompted to produce actions, HYPOTHESIZER is provided with (a) the current observation from the environment, (b) the last 3 actions it took (and their respective explanations and notes), (c) it's current working memory, (d) a list of possible actions and which arguments they take. All output is produced as a JSON dictionary.

The working memory that Hypothesizer stores can be long, especially for longer tasks. An example of the Hypothesizer agent's working memory at the successful completion of a specific shorter task (*Proteomics, Normal, Seed 2*) is provided below.

---

```
Task Description: You are in a biological preserve on Planet X, that has 5 different animal species.
We suspect that one of these animal species is not native to the area, but migrated from an isolated
island in the recent past. Your task is to use the proteomics meter to analyze the proteins of each
of the 5 animal species, which can be found throughout the environment away from the central statue
area, and determine which species is the anomaly. Once you have completed your task, return to the
statue area and drop the red flag directly beside the statue of the animal species that is the
anomoly.

"working_memory": {
    "scientific_knowledge": [
        {
            "measurement": "The spheroid (uuid: 23195) is located in Possible Animal Area 1.",
            "step": 2
        },
        {
            "measurement": "The proteomics meter (uuid: 33276) is now in my inventory.",
            "step": 8
        },
        {
```

```
                "measurement": "The proteomics meter (uuid: 33276) requires two argument objects to be
                    used.",
                "step": 10
        },
        {

                "hypothesis": "if (object:spheroid) is (action:directly in front) then
                    (object:proteomics meter) can be used to analyze its proteins.",
                "status": "confirmed",
                "step": 13,
                "supporting evidence": "Using the proteomics meter on the spheroid yielded protein data:
                    Protein A: 0.87, Protein B: 0.80."
        },
        {

                "measurement": "The spheroid (uuid: 23195) has Protein A level of 0.87 and Protein B
                    level of 0.80 when analyzed using the proteomics meter.",
                "step": 13
        },
        {

                "measurement": "The animaplant (uuid: 20030) is located in Possible Animal Area 2.",
                "step": 14
        },
        {

                "hypothesis": "if (object:animaplant) is (action:analyzed) using (object:proteomics
                    meter) then (property:protein data) will be obtained.",
                "status": "confirmed",
                "step": 15,
                "supporting evidence": "Using the proteomics meter on the animaplant yielded protein
                    data: Protein A: 0.76, Protein B: 0.67."
        },
        {

                "measurement": "The animaplant (uuid: 20030) has Protein A level of 0.76 and Protein B
                    level of 0.67 when analyzed using the proteomics meter.",
                "step": 15
        },
        {

                "measurement": "The animal in Possible Animal Area 3 (uuid: 20031) is not accessible.",
                "step": 17
        },
        {

                "hypothesis": "if (object:animal) is (action:teleported to) using (object:uuid: 20031)
                    then (property:accessibility) will be false.",
                "status": "confirmed",
                "step": 22,
                "supporting evidence": "Attempting to teleport to the animal object with UUID 20031
                    resulted in an error indicating that no object with that UUID was found."
        },
        {

                "measurement": "There is a tree (big) trunk blocking the path to the west.",
                "step": 25
        },
        {

                "measurement": "I successfully moved south to navigate around the tree (big) trunk.",
                "step": 26
        },
        {

                "measurement": "I successfully moved west to get directly in front of the pink
                    animal.",
                "step": 28
        },
        {

                "measurement": "I rotated to face west, aligning myself with the pink animal.",
                "step": 30
        },
        {

                "measurement": "The object directly in front of me, which I thought was a pink
                    animal, is actually grass (uuid: 20981).",
                "step": 31
        },
        {

                "measurement": "The proteomics meter (uuid: 33276) was used to investigate the grass
                    (uuid: 20981), but the results were inconclusive.",
                "step": 31
        },
        {

                "measurement": "The object with UUID 44261 is a plant, not an animal.",
                "step": 37
        },
        {

                "measurement": "The proteomics meter (uuid: 33276) was used to investigate the
                    animaplant (uuid: 44261). The results are as follows:
                    Protein A: 0.76, Protein B: 0.67.",
```

```
        "step": 37
    },
    {
        "measurement": "The object with UUID 12545 is a plant, not an animal.",
        "step": 38
    },
    {
        "measurement": "The proteomics meter (uuid: 33276) was used to investigate the plant
            (uuid: 12545), but the results were inconclusive.",
        "step": 38
    },
    {
        "measurement": "The vortisquid (uuid: 41429) is one of the animal species in the
            biological preserve.",
        "step": 149
    },
    {
        "hypothesis": "if (object:vortisquid) is (action:analyzed) using
            (object:proteomics meter) then (property:protein data) will be obtained.",
        "status": "confirmed",
        "step": 150,
        "supporting evidence": "Using the proteomics meter on the vortisquid yielded
            protein data: Protein A: 0.38, Protein B: 0.46."
    },
    {
        "measurement": "The vortisquid (uuid: 41429) has Protein A level of 0.38 and
            Protein B level of 0.46 when analyzed using the proteomics meter.",
        "step": 150
    },
    {
        "measurement": "The prismatic beast (uuid: 25282) is located in Possible Animal
            Area 5.",
        "step": 151
    },
    {
        "hypothesis": "if (object:prismatic beast) is (action:analyzed) using
            (object:proteomics meter) then (property:protein data) will be obtained.",
        "status": "confirmed",
        "step": 153,
        "supporting evidence": "Using the proteomics meter on the prismatic beast
            yielded protein data: Protein A: 0.73, Protein B: 0.68."
    },
    {
        "measurement": "The prismatic beast (uuid: 25282) has Protein A level of 0.73
            and Protein B level of 0.68 when analyzed using the proteomics meter.",
        "step": 153
    },
    {
        "measurement": "The prismatic beast (uuid: 5059) is located in Possible Animal
            Area 7.",
        "step": 156
    },
    {
        "measurement": "The prismatic beast (uuid: 5059) has Protein A level of 0.73 and
            Protein B level of 0.68 when analyzed using the proteomics meter.",
        "step": 157
    },
    {
        "measurement": "The echojelly (uuid: 4429) is one of the animal species in the
            biological preserve.",
        "step": 160
    },
    {
        "hypothesis": "if (object:echojelly) is (action:analyzed) using
            (object:proteomics meter) then (property:protein data) will be obtained.",
        "status": "confirmed",
        "step": 161,
        "supporting evidence": "Using the proteomics meter on the echojelly yielded
            protein data: Protein A: 0.86, Protein B: 0.82."
    },
    {
        "measurement": "The echojelly (uuid: 4429) has Protein A level of 0.86 and
            Protein B level of 0.82 when analyzed using the proteomics meter.",
        "step": 161
    },
    {
        "measurement": "The vortisquid (uuid: 41429) has significantly different protein
            levels (Protein A: 0.38, Protein B: 0.46) compared to the other species,
            indicating it is the anomaly.",
        "step": 162
    },
```

```
{
    "measurement": "The object with UUID 33979 (red flag) is not directly in front of
        me or in my inventory.",
    "step": 172
},
{
    "hypothesis": "if (object:red flag) is (action:teleported to) using
        (object:uuid: 33979) then (property:accessibility) will be true.",
    "status": "pending",
    "step": 172,
    "supporting evidence": "Attempting to pick up the red flag from the Instrument
        Table resulted in errors indicating that the object with UUID 33979 could
        not be found."
},
{
    "measurement": "The object with UUID 33979 (red flag) is not directly in front
        of me or in my inventory.",
    "step": 173
},
{
    "hypothesis": "if (object:red flag) is (action:teleported to) using
        (object:uuid: 33979) then (property:accessibility) will be true.",
    "status": "pending",
    "step": 173,
    "supporting evidence": "Attempting to pick up the red flag from the Instrument
        Table resulted in errors indicating that the object with UUID 33979 could
        not be found."
},
{
    "measurement": "The object with UUID 33979 (red flag) is located at
        coordinates (15, 20).",
    "step": 174
},
{
    "hypothesis": "if (object:red flag) is (action:teleported to) using
        (object:uuid: 33979) then (property:accessibility) will be true.",
    "status": "confirmed",
    "step": 174,
    "supporting evidence": "Successfully teleported to the object with
        UUID 33979 (red flag) at coordinates (15, 20)."
},
{
    "measurement": "The object with UUID 33979 (red flag) is now in my inventory.",
    "step": 175
},
{
    "hypothesis": "if (object:red flag) is (action:picked up) then
        (property:accessibility) will be true.",
    "status": "confirmed",
    "step": 175,
    "supporting evidence": "Successfully picked up the red flag after teleporting
        to it."
},
{
    "measurement": "The object with UUID 33979 (red flag) is located at
        coordinates (15, 20).",
    "step": 176
},
{
    "hypothesis": "if (object:red flag) is (action:teleported to) using
        (object:uuid: 33979) then (property:accessibility) will be true.",
    "status": "confirmed",
    "step": 176,
    "supporting evidence": "Successfully teleported to the object with
        UUID 33979 (red flag) at coordinates (15, 20)."
},
{
    "measurement": "The object with UUID 33979 (red flag) is now in my inventory.",
    "step": 176
},
{
    "hypothesis": "if (object:red flag) is (action:picked up) then
        (property:accessibility) will be true.",
    "status": "confirmed",
    "step": 176,
    "supporting evidence": "Successfully picked up the red flag after teleporting to it."
},
{
    "measurement": "I teleported to the Statue of a vortisquid at
        coordinates (18, 17).",
    "step": 176
```

```
        }
    ]
}
```

---

## C.4 Cost analysis

An approximate cost analysis of each baselines model is provided below, in Table 10. Model cost varies across task, environment complexity, and the number of items stored within a model's memory (e.g. the size of the history in REACT, or the size of the memory in HYPOTHESIZER), so approximate averages are shown.

The full benchmark contains $8$ discovery themes $\times$ 3 difficulty levels $\times$ 5 seeds $= 120$ tasks to evaluate. *Easy* tasks are evaluated to 100 steps, while *Normal* and *Challenge* tasks are evaluated to 1000 steps or a hard \$125 limit, whichever came first. This results in a total upper-bound estimate of $40 \times 100(easy) + 40 \times 1000(normal) + 40 \times 1000(challenge) = 84,000$ steps required to complete an evaluation run, assuming no agents finish early due to task completion.

Table 10: Approximate cost estimate for the models investigated in this work (all using GPT-4o). **Assumes a cost of \$5/M input tokens, and \$15/M output tokens** (which was pricing as of submission time; current pricing as of acceptance is approximately half this, or \$2.50/M input tokens, and \$10/M output tokens).

| Model | Cost per 100 steps | Total cost estimate (120 tasks, 84k steps) |
|---|---|---|
| REACT | \$2-4 / 100 steps | \$3,360 |
| PLAN + EXECUTE | \$3-4 / 100 steps | \$3,360 |
| HYPOTHESIZER | \$10 / 100 steps | \$8,400 |

## D  Automatic Evaluation of Explanatory Discovery Knowledge

The following prompt is used to automatically evaluate explanatory discovery knowledge, in reference to gold reference questions provided automatically by the DISCOVERYWORLD task scorer:

---

```
Your task is to evaluate the knowledge of an agent playing a game about
scientific discovery, to see if they have figured out the discovery
required to solve the task. I will supply you with: (1) the task
description, (2) the agent's knowledge, and (3) a single critical question
about the discovery, that you should answer with either true (1) or
false (0).

Task Description:
'''
[*INSERT TASK DESCRIPTION HERE*]
'''

Agent's Knowledge:
'''
[*INSERT AGENT'S KNOWLEDGE HERE*]
'''

Critical Question:
'''
[*INSERT SINGLE KNOWLEDGE EVALUATION QUESTION HERE FROM SCORECARD*]
'''

Please answer this question by responding '1' if the agent's knowledege
reflects having discovered the information in the critical question, and
'0' otherwise.  This response should be in the 'evaluation' key of the
response. The response format is a JSON dictionary containing three keys:
```

`criticalQuestion`, `evaluation`, and `explanation`.
```
{
    \"criticalQuestion\": \"repeat the critical question\",
    \"evaluation\": 0 or 1 (as integers),
    \"explanation\": \"provide a brief explanation for evaluation, making
        reference to the agent's knowledge and whether or not it reflects
        the critical question.\"
}
```

---

The following is an example of the scorer producing a **positive assessment**:

---

```
Task: Proteomics

"evaluation": [
    {
        "criticalQuestion": "Does it clearly state that the vortisquid has
                proteomics values that appear to be outliers compared to the
                other animals, whose proteomics values all cluster together?",
        "evaluation": 1,
        "explanation": "The agent's knowledge includes a specific
                measurement stating that the vortisquid has significantly
                different protein levels (Protein A: 0.38, Protein B: 0.46)
                compared to the other species, indicating it is the anomaly.
                This directly addresses the critical question by identifying
                the vortisquid's proteomics values as outliers compared to
                the other animals."
    }
],
"evaluation_totalscore_raw": 1,
"evaluation_totalscore": 1.0,
```

---

The following is an example of the scorer producing a **negative assessment**:

---

```
Task: Translation

"evaluation": [
    {
        "criticalQuestion": "Does it clearly state that: The word 'Womple'
                means something like 'bring me'?",
        "evaluation": 0,
        "explanation": "The agent's knowledge does not explicitly state
                that the word 'Womple' means 'bring me'. The agent has
                recorded instances where the elder and the trainer used the
                word 'Womple', but there is no clear interpretation or
                translation of the word provided in the knowledge. The agent
                has noted that the elder seems interested in items in the
                inventory and asked if the agent wants to give the items to
                him, but this does not directly translate 'Womple' to
                'bring me'."
    },
    {
        "criticalQuestion": "Does it clearly state that: The word
                'florpt' means 'flower'?",
        "evaluation": 0,
        "explanation": "The agent's knowledge does not explicitly state
                that the word 'florpt' means 'flower'. While there are
                measurements and interactions involving the word 'florpt'
```

```
                and objects like flowers, there is no direct confirmation
                or hypothesis that equates 'florpt' to 'flower'."
    }
],
"evaluation_totalscore_raw": 0,
"evaluation_totalscore": 0.0,
```

---

# E   Human Scientist Participants

**General Details**   To compare model performance against human performance, we recruited 11 practicing human scientists to complete the DISCOVERYWORLD tasks, with their performance shown in Table 6. Scientists were recruited on the UPWORK platform, each with: (1) an MSC or PHD in a natural science, (2) self-evaluated comfort and fluency with statistical methods and common software like spreadsheets, (3) comfort and previous experience with 2D top-down games. To evaluate the latter, all participants were required to complete a screening task involving a *Tutorial* in DISCOVERYWORLD. Participants were given a maximum of one hour to complete each task. While our agent models are run in a zero-shot setting, humans can't forget their past experiences, so our participants only completed the *normal* and *challenge* settings, to prevent any easily-discovered knowledge in the *easy* setting from providing an advantage in other settings. To collect accurate estimates of task difficulty, all humans completed the same seed (SEED 0) of each task. To evaluate discovered knowledge, participants were asked to keep notes, and explicitly state their hypotheses, and supporting evidence. Due to the varied nature of these notes (some wrote a few words, others entire pages per task) and their modality (text, spreadsheets), discovery knowledge was evaluated manually using the following criteria: 1 (discovered all discovery knowledge in scorecard), 0.5 (discovered some discovery knowledge in scorecard), or 0 (did not find critical discovery knowledge). Other metrics *(task completion, procedural knowledge)* were evaluated automatically. At their option, not all participants completed all tasks.

**Informed Consent**   Details about the project's objective and the intended use of data generated during game play was provided to participants in the initial job description. Participants were further provided with a participation agreement detailing that all data generated from the project will be owned by AI2, which they agreed to by choosing to participate in the project.

**Participant Instructions**   Participant instructions can be found in README-USERSTUDY.MD on the DISCOVERYWORLD code repository.

**Potential Participant Risks**   To protect the personal information and anonymity of the human scientists we recruited, no personally identifying information was collected and all data is anonymized.

**Recruitment**   Participants were recruited and hired on Upwork, where they were initially required to submit a proposal detailing their educational background, experience with statistical analyses and Python programming, and interest in video games.

**Compensation**   Human scientists were compensated at a rate ranging from USD\$20-\$30/hr for their work, based on the rate they bid in their initial job proposals. The total amount spent on participant compensation did not exceed \$4,000.

**Participants' Expertise**   The participants we recruited had Master's or Doctorate degrees in a range of disciplines. Their educational and/or work experience spanned the following fields: astrophysics, bioinformatics, biology, biomedical engineering, biostatistics, chemistry, computer engineering, data science, electrical engineering, geology, machine learning, mathematics, and physics.

**Graphical Interface**   Humans made use of the 2D graphical user interface for their study, rather than a text-only version (as some agent models use). Figure 4 shows an example of the interface.

**Time limit**   Human participants were given a maximum of 1 hour to complete a given task.

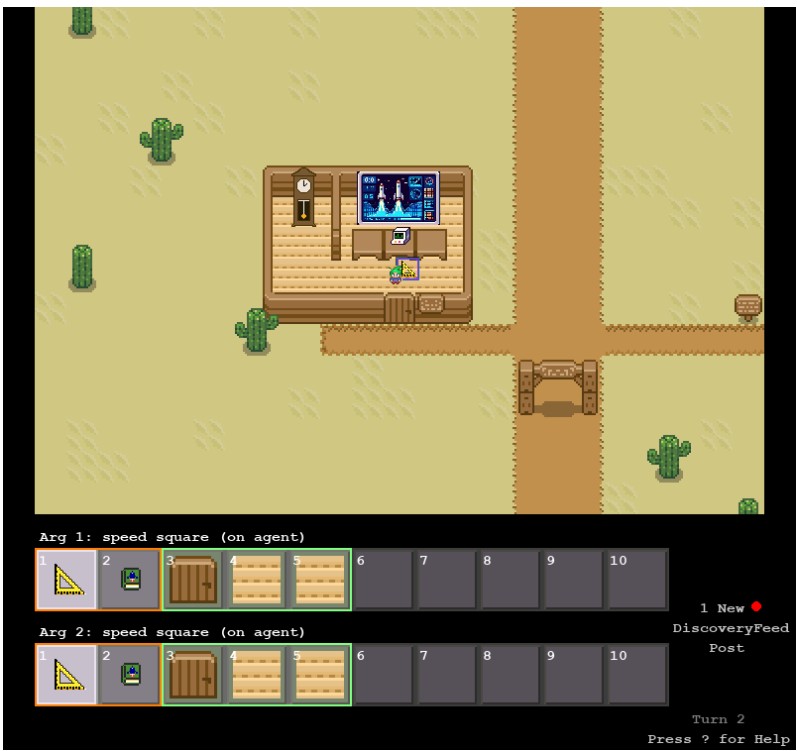

Figure 4: Example of the user interface the participants used. This is for the *It's (not) Rocket Science!* theme.

**Zero-shot Performance** While participants were allowed to retry tasks that they did not complete successfully the first time, we did not include any retries in our evaluation of human performance, to give an accurate analog of *zero-shot* model performance.

**Data** Instructions on how to get the anonymized data from human participants is available at https://github.com/allenai/discoveryworld.

**Save failures** DISCOVERYWORLD automatically saves log files to evaluate performance. Some participants noted infrequent issues with this automatic save on the WINDOWS platform that we were unable to resolve. As such, a small amount of data was lost, and is not included in this analysis.

