# DISCOVERYWORLD: A Virtual Environment for Developing and Evaluating Automated Scientific Discovery Agents

**Peter Jansen**[*‡], **Marc-Alexandre Côté**[†], **Tushar Khot**[*] **Erin Bransom**[*], **Bhavana Dalvi Mishra**[*],
**Bodhisattwa Prasad Majumder**[*], **Oyvind Tafjord**[*], **Peter Clark**[*]
[*]Allen Institute for Artificial Intelligence    [†]Microsoft Research    [‡]University of Arizona
peterj@allenai.org

## Abstract

This is the supplementary checklist for DISCOVERYWORLD. All our code, data, and other supplementary content is available at:
`https://github.com/allenai/discoveryworld` .

## 1 Supplementary Information

1. **Dataset or Benchmark:** Is this a dataset or a benchmark? **A benchmark**

2. **Benchmark:** For benchmarks, the supplementary materials must ensure that all results are easily reproducible (i.e. all necessary datasets, code, and evaluation procedures must be accessible and documented)

   **Code (Benchmark):** DISCOVERYWORLD is released as a PIP-installable library. It can be installed in a few minutes by following the instructions at `https://github.com/allenai/discoveryworld` .

   **Code (Baseline Models):** All agent baseline models (as well as an additional random baseline model) are provided in the repository, at: `https://github.com/allenai/discoveryworld/tree/main/agents` . The agents include detailed instructions, to get up and running quickly. Additional templates are provided to help users build their own agents quickly and easily.

   **Evaluation Procedures:** All the evaluation procedures are documented both in the paper, and the repository. Our evaluation procedures are straightforward: Each {DISCOVERY THEME, DIFFICULTY} combination (of which there are 24) are run for each of 5 parametric seeds (seeds 0, 1, 2, 3, 4). The results are averaged across seeds to arrive at the final performance shown in each table. We include analysis scripts that automatically perform this averaging for the user, and describe their use in the repository at: `https://github.com/allenai/discoveryworld/tree/main/data` .

   **Data:** Our contribution is a benchmark, not a dataset, but we do release the supplementary data produced by our benchmark and used for follow-on analyses in the paper, to support replicability. This is described below.

   **Data (Agents):** We release all data produced by the agents reported in this paper, to support both replicability and follow-on analyses. Links to the agent data (in a public GOOGLE DRIVE folder), as well as details on the dataset formats, and how to use the data with the included analysis scripts, is provided here: `https://github.com/allenai/discoveryworld/tree/main/data` .

   **Data (Human Scientist Players):** We release all data produced by the human scientists who provided a human baseline for the DISCOVERYWORLD benchmark. This centrally

includes (1) the detailed playlogs, i.e. what each human did at each step of each game they played, and (2) any notes that the human scientists took while playing. These notes have been sanitized of any identifying information, and converted to easily-parsable formats (i.e. TXT, TSV, PYTHON NOTEBOOKS).

3. **Accessibility**: The following are accessibility items on the submission checklist:

**Links to access the benchmark:** The link to access the benchmark is provided in the abstract of the main submission ( `https://github.com/allenai/discoveryworld/tree/main/data` ).

**Any data should use open and widely used formats. Simulation environments should explain how they can be used:** Our data are stored in widely accessible standard formats (e.g. JSON, TXT, TSV), with supporting documentation detailing their formats: `https://github.com/allenai/discoveryworld/tree/main/data` . Our simulation environment for our benchmark includes detailed API documentation on the main page of the repository, and example code to help users get new agents running quickly ( `https://github.com/allenai/discoveryworld/tree/main/` ).

**Long-term preservation.** Code and data are provided on GITHUB with GOOGLE DRIVE links for large files. All aspects will be publicly available for a long term.

**Explicit Licence:** Our benchmark is licensed using APACHE 2.0, which is included in the GITHUB repository.

**Structured Metadata for a dataset:** This is not applicable (DISCOVERYWORLD is a benchmark that takes place in a virtual environment, and there is no typical data release as such). The associated data from our baseline runs is included as links to Google Drive folders on the repository ( `https://github.com/allenai/discoveryworld/tree/main/data` ).

**A persistent dereferenceable identifier (e.g. a code repository such as GitHub):** The repository for our benchmark is: `https://github.com/allenai/discoveryworld` .