# OpenReview forum: "DiscoveryWorld: A Virtual Environment for Developing and Evaluating Automated Scientific Discovery Agents"
_NeurIPS.cc/2024/Datasets_and_Benchmarks_Track — NeurIPS 2024 Track Datasets and Benchmarks Spotlight_

### Official Review · Reviewer_BBUY · 2024-06-23

**Rating:** 7
**Confidence:** 4

**Review:**

# quality
Good
- Provides a wide variety of test domains that challenge general scientific reasoning capabilities in different ways.
- Weakness: conflates evaluation of reasoning with basic agent navigation and manipulation skills. These baseline requirements to navigate or manipulate the environment are too abstract to be of high value, but are concrete enough in the current implementation to hinder agent performance.

# clarity
Good
- Describes the evaluation criteria in detail and information on the agent observation spaces.

# originality
Modest
- There are few benchmarks for scientific reasoning (knowledge discovery) tasks.
- The technical components are not novel per se, but their target application is somewhat unique.

# significance
Modest
- The benchmark is well-suited to supporting investigation of general methods for scientific knowledge discovery.
- But success in this benchmark is unlikely to directly translate into domain-specific agents that address scientific discovery in real scientific tasks. The abstraction here may prove to hinder the ability to work on the hardest parts of discovery tasks in "real" scientific domains.
- Will be of interest to researchers broadly interested in autonomous agents, LLMs, and AI applications to science.

**Strengths:**

The key benchmark strength is conforming many scientific knowledge discovery tasks to a single broader interface and providing a harness for evaluating agent success at those tasks. Including results from trained human experts offers a clear baseline for evaluating progress.

Many agent (LLM) benchmarks have focused on knowledge possessed by models and reasoning in areas like mathematics or programming. Having a comparable benchmark for scientific knowledge discovery will provide a new axis of agent capabilities of broad interest.

There are many positive societal implications for improving the ability of agents to produce scientific discoveries. Negative implications are covered.

**Additional Feedback:**

- Please explain how the API diverges from the standard gym API (if at all).
	- Can arbitrary RL agents be plugged in? Or is the action space a problem?
	- Can the environment be parallelized or accelerated?
- Why is the environment "embodied" in the sense of the agent needing to navigate and visit rooms?
	- This seems to impede agent success but is (largely, but not wholly) independent to the scientific experimentation and reasoning elements being tested for autonomous discovery
	- The fact that evaluations were cleaner after giving the agent the DiscoveryFeed and enabling agent teleportation suggests this is a more reasonable focus for the benchmark.
	- The combination of task types seems to muddy the benchmark: LLM reasoning about navigation is not particularly interesting when considering agents capable of general scientific discovery. to the degree it is the interest part becomes manipulation of experimental equipment, which is a different level of abstraction than the benchmark aims for.
	- It seems plausible to alter the benchmark to provide a graph-based abstraction of locations (that is, teleportation) while retaining the important discovery challenges.
- What is varied when changing task difficulty?
	- The supplement provides details, but it would help to reference an example in section 3.2.
- Table 5
	- "Explanatory knowledge discovery" is not reported (the caption claims 3 metrics but 2 are shown).
- Table 6
	- Report human averages for normal and challenge tasks. The values are mentioned in the text and this would create parity with the Table 5 structure.
	- If possible (due to space) it would help readability to provide a bar plot showing human vs agent performance divided by topic and task and including a report of variation in results (like standard error or inter-quartile range).
- Any idea why ReACT has the highest completion rates?
	- Are the results simply high variance?
	- Or is there something systemic about how ReACT handles the task that helps? Does the long trajectory history provide useful clues rather than distracting on these tasks?

**Clarity:**

Yes. The paper articulates the need for an (abstracted) general benchmark for scientific reasoning to test agent capabilities in autonomous discovery. Design decisions, experiments, and results are (generally) clearly described. Some minor notes below on nit-picks.

**Correctness:**

Yes. Experiments demonstrate that several reasonable baseline agents vary widely in performance on the tasks, particularly at higher difficulties. Human experts (holding a masters of PhD in the natural sciences) can solve most of the tasks, but struggle with the hardest. The tests also partially isolate aspects of the benchmark not related to the core reasoning and experimentation process in unit tests of navigation and object manipulation.

**Documentation:**

Detailed reproducibility instructions and code are provided (in the supplement). Human study participant ethical details are included.

**Ethics:**

No.

**Limitations:**

Captures key limitations of abstraction, cost, and societal impact of dual use discoveries.

A somewhat meta point to consider (not necessary for the text): will automated discovery induce under-provision of the public good of science (scientific discoveries or knowledge)?
To the degree that automated discovery can yield valuable knowledge without the need of human peer review for validation, a group could monopolize automated discovery systems to gather scientific knowledge while refraining from publicly sharing these results. This is a long-range potential impact of automated discovery work: not one necessarily of immediate or obvious grave importance, but something to consider.

**Opportunities For Improvement:**

The environment requirements for navigation distract from the core tasks of the benchmark and clearly have a negative impact on agent performance that is largely orthogonal to scientific discovery capabilities. The benchmark would benefit from a simplified API that removed most of the navigation and manipulation capabilities.

More detailed comments in the suggestions section.

**Relation To Prior Work:**

Yes. The benchmark is unique in including a variety of domains (8 total) with a shared interface and scoring system.

**Summary And Contributions:**

The paper proposes an 2D top-down navigation environment that hosts a suite of procedurally generated tasks requiring agents to obtain scientific knowledge in 8 domains. Procedural generation uses templates to vary task difficulty and provide new instances. The benchmark includes tests for task completion, task steps, and final explanation (answer) evaluation. A set of unit tests isolate baseline agent skills to navigate around and manipulate objects in the environment.

A series of evaluations shows a range of model performances in the tasks and demonstrates the difficulty of the hardest tasks for trained natural scientists.

---

> ### Author Rebuttal · Authors · 2024-08-16
>
> We’d like to thank the reviewer for their kind and thoughtful comments, and we’re glad you enjoyed the work, and for the overall positive comments and rating. We hope to address the constructive feedback comments here.
>
> - Thanks for noting that DiscoveryWorld provides a wide variety of tasks that challenge general scientific reasoning capabilities, that this benchmark is well-suited to investigating general methods of automated scientific knowledge discovery, and the many positive societal implications for improving agents' abilities to produce scientific discoveries.
> - Thank you for highlighting the novel contributions to evaluation, and the large differences in performance between current agents and real human natural scientists.
>
>
> **Response to Questions:**
>
> **Scientific Discovery skills vs Common-sense/World Knowledge skills:** "The environment requirements for navigation distract from the core tasks of the benchmark and clearly have a negative impact on agent performance that is largely orthogonal to scientific discovery capabilities. The benchmark would benefit from a simplified API that removed most of the navigation and manipulation capabilities."
>
> **Response:** This is a great question that we ourselves considered a great deal when designing DiscoveryWorld.  Scientific discovery is usually divided into sub-areas (e.g. data-driven discovery, literature-driven discovery, experiment-driven discovery), and DiscoveryWorld was tailored for experiment-driven discovery, which involves agents formulating hypotheses and conducting iterative experiments. This process includes designing and running experiments, analyzing outcomes, and refining hypotheses based on the results. This process of running experiments does conventionally require some world knowledge, such as searching, sample collection, and object manipulation—areas where agent models have traditionally struggled.
>
> We built DiscoveryWorld to measure an agent's ability to solve this end-to-end, but also included optional simplifications:
>
> 1) Teleport-to-location action: Agents can teleport to any of a set of named locations (hand-picked to be critical for a given task).  The agents frequently use these.
>
> 2) Teleport-to-object action: Agents can teleport to any object that it has seen before, by name (i.e. teleport to stove).  The agents sometimes seem to have figured out a sort of 'fast travel' by first teleporting to an object 2-3 steps away, and if they don't see what they're looking for, they teleport to another object 2-3 steps away, saving themselves the walk.
>
> 3) "Easy" tasks: Among other simplifications, they greatly simplify the navigation/manipulation challenges involved in making a discovery.
>
> 4) Unit tests: DiscoveryWorld has 10 unit tests to evaluate an agent's basic skills in navigation, manipulation, and interaction with the environment. The Hypothesizer agent was able to successfully complete 6 out of these 10 tests, particularly excelling in object manipulation and partially in navigation. This performance indicates that, at least in certain scenarios, the abilities of navigating and manipulating objects may not be significant obstacles to completing tasks successfully.
>
> 5) Transfer: In an ideal world, if an agent performs well at DiscoveryWorld, someone might rightly want to transfer it into a robotic model to work in a real lab on a specific kind of problem.  In those settings, navigation and environment manipulation skills are required (and essential).
>
> **How does the API diverge from the standard Gym API.  Can Arbitrary RL agents be plugged in, or is the action space a problem:**
>
> The API closely resembles the Gym API where the observation is multi-modal: both the text and the visual frame describe the agent’s surroundings. As such, most existing text game agents (many of which are RL text agents) should be able to be plugged in with minimal effort (i.e., by ignoring visual modality).
>
> To the broader question of whether arbitrary RL agents can be plugged in: This is a broader research methods question that's been addressed extensively in the text-game literature, due to the mismatch in what the action spaces of text games (often combinatorial) tend to look like compared to conventional RL problems (e.g. game controller buttons). There are ways of flattening the action space through "valid action handicap" (i.e. creating a list of valid actions for agents to select from/rank/etc), that our API also supports.
>
> **What is varied when changing task difficulty? The supplement provides details, but it would help to reference an example in section 3.2.**
>
> We will make this clearer in the paper and on the companion website through examples.  Across different difficulty settings, there are always large changes in both the complexity of the problem and the environment itself.  For example, the "easy" problems are typically performed in very small/constrained environments (like single rooms with few/no distractors, similar to the Unit Test environments), have much simpler problems, and may have their answering mechanism changed to a forced-choice task (e.g. answering by moving a specific kind of object to a container).  The harder tasks have far richer environments, far more complex discoveries, and in many cases are much more open ended.
>
>
> **Any idea why ReACT has the highest completion rates? … Does the long trajectory history provide useful clues rather than distracting on these tasks?**
>
> We have broad hypotheses -- e.g. as you say, ReACT tends to have a very long trajectory history that it preserves, whereas the Hypothesizer agent has a much more complex abstractive memory that is perhaps better for inference and explanation, but poor at maintaining exact records of past steps, which might be important for some tasks.  The different styles of memory seem to serve ReACT better in some tasks, and Hypothesizer better in others.
>
>
> **Minor:**
> Thanks for the layout comments on Tables 5/6 – we will fix these issues.

---

> > ### Comment · Reviewer_BBUY · 2024-08-17
> >
> > Thanks for the replies!
> >
> > One minor comment:
> >
> > >Transfer: In an ideal world, if an agent performs well at DiscoveryWorld, someone might rightly want to transfer it into a robotic model to work in a real lab on a specific kind of problem. In those settings, navigation and environment manipulation skills are required (and essential).
> >
> > I assumed this was the intent. But given that robot embodiments vary widely and come with their own set of challenges it does not seem like asking an agent to walk to rooms would resemble what the transfer domain of automated lab robots would require.
> >
> > All this is to say I'd love a "simple" environment version (or config file) that toggled all the movement / embodiment requirements off. The ugly part there seems to be the coupling of movement with task difficulty in some cases.

---

> > > ### Author Rebuttal · Authors · 2024-08-19
> > >
> > > Thanks -- we share your interest in this important question.  While we're interested in the "let's make the task more realistic by adding in more complexity" story, we're also very interested in the "can the separate elements of the task be distilled even further" ablation-style research methods questions.  We've added this request for having a setting that removes all navigation and object manipulation into the official Github repository as a feature request, to discuss how it might best be implemented, and see if we can add this in.
> > >
> > > (There are pragmatic considerations that make implementing this non-trivial -- e.g. some of the tasks require a fair amount of spatial reasoning, such as some of the difficulties of the Rocket Science task, which require using different spatial locations on the map to measure the planetary diameter using Eratosthenes' method).  We also can't easily remove navigation or object manipulation from the simulator itself through a setting (navigation and object manipulation are at the heart of the "Object Tree" model of word simulation), but we're trying to figure out if there's a way of adding this through different game modes (e.g. if a setting made the agent omniscient, where it can always observe all objects in the map, and perform actions (e.g. using a thermometer on a specific object) without being near them/having first picked them up/etc.)  The benefit of this is that there'd be no search/navigation/minimal object manipulation. One of the challenges is that the agent would then suddenly be observing a lot more objects than just its immediate surroundings, and there are regularly on the order of 1000+ objects in the world, so this would place an added burden on the agent to be able to handle having a list of all the objects in the simulator in its prompt simultaneously -- it's not clear if we'd have to make a new set of scenarios to support this faithfully/tractably.  All this is to say that we share your interest in having this kind of ablation naturally supported, and well figure out if there's an elegant way of adding this to the next major subversion.

---

> > > > ### Comment · Reviewer_BBUY · 2024-08-21
> > > >
> > > > Thank you for adding the request and explaining the technical challenges. Adapting a game to new modes is not a trivial task and I understand the nuanced difficulties.

---

### Official Review · Reviewer_TdaW · 2024-07-24
**Review for DiscoveryWorld**

**Rating:** 7
**Confidence:** 4
**Correctness:** Yes
**Clarity:** Yes

**Review:**

The paper is clearly written with details on most parts of the process.

Pros and cons are discussed in strengths and weaknesses below.

**Strengths:**

Interesting Topic: The creation of a virtual environment for scientific discovery is a highly relevant and intriguing topic, especially as AI continues to advance into more complex and nuanced fields.

Wide Coverage of Real-World Scenarios: DiscoveryWorld covers a broad spectrum of scientific domains, encouraging the development of discovery skills. This variety ensures that agents need to apply a wide range of knowledge and techniques.

**Additional Feedback:**

Future Works on the Environment: What are your opinions on the future directions for developing DiscoveryWorld? Should future enhancements rely solely on large language models (LLMs), or should there be an integration of other methods such as planning and reinforcement learning? How do you envision these methods complementing each other in improving the capabilities of scientific discovery agents?

LLM Performance and Real Contribution: If a large language model in the future achieves excellent performance on the tasks presented in DiscoveryWorld, what would you consider its real contribution? Would it be more valuable if an LLM with a vast amount of prior knowledge excels in this environment, or if a model performs well on tasks without extensive prior knowledge, akin to the challenges in IVRE? Or in other words, which of the following do you consider to be more contributing to the field, a model relying on a large amount of prior knowledge performs well on a task requiring prior knowledge, or a model without much prior knowledge develops some sense of general problem-solving for a non-practical task?

**Documentation:**

Yes

**Limitations:**

Yes

**Opportunities For Improvement:**

Lack of Experiments Using Reinforcement Learning and Planning: The paper does not provide experiments or evaluations using reinforcement learning (RL) or other planning-based approaches. Considering RL's prominence in solving complex tasks through iterative learning and adaptation, its absence is a notable gap. Incorporating RL experiments could provide deeper insights into the potential and limitations of DiscoveryWorld.

**Relation To Prior Work:**

Yes

**Summary And Contributions:**

The paper introduces DiscoveryWorld, a virtual environment specifically designed for developing and benchmarking AI agents' capabilities in end-to-end scientific discovery. DiscoveryWorld offers a range of challenge tasks covering diverse topics like radioisotope dating, rocket science, and proteomics. These tasks require agents to form hypotheses, design and execute experiments, analyze results, and draw conclusions. The environment is text-based with an optional 2D visual overlay, containing 120 different tasks across eight topics, each with three levels of difficulty. The paper highlights the novel challenges presented by DiscoveryWorld, illustrating that strong baseline agents struggle with most tasks, thereby emphasizing the potential of DiscoveryWorld in advancing AI scientific discovery agents.

---

> ### Author Rebuttal · Authors · 2024-08-16
>
> We’d like to thank the reviewer for their kind and thoughtful comments, and we’re glad you enjoyed the work, and for the overall positive comments and rating. We hope to address the constructive feedback comments here.
>
> - Thanks for your comments that the paper is well-written, that it highlights the novel challenges presented in DiscoveryWorld, and for emphasizing the potential of DiscoveryWorld to advance AI scientific discovery agents.
> - Thanks also for noting that DiscoveryWorld is an environment with broad coverage of real-world scientific discovery scenarios, that encourages the development of discovery skills, and in applying a wide range of knowledge and techniques.
>
>
> **Responses to Questions:**
>
> **Lack of Experiments Using RL and Planning:** "The paper does not provide experiments or evaluations using reinforcement learning (RL) or other planning-based approaches. Considering RL's prominence in solving complex tasks through iterative learning and adaptation, its absence is a notable gap. Incorporating RL experiments could provide deeper insights into the potential and limitations of DiscoveryWorld."
>
> **Response:** Thanks for this comment:
> - *Planning:* We don't yet include any formal planning approaches (e.g. PDDL) because it's been shown that building the domain file from exploring the environment is extremely challenging (e.g. Zhang et al. (2024) PDDLEGO: Iterative Planning in Textual Environments), and still unsolved on much simpler environments. As a small step towards this goal, we do include an LLM-planning agent baseline (Plan+Execute), where the agent model is used to generate a plan, then execute it.
>
> - *Reinforcement Learning:* In terms of pure RL approaches, we would argue that these still struggle on complex long-horizon tasks with sparse reward, such as DiscoveryWorld.  That being said, we attempted to apply a number of baselines during our model development phase, but our pilot experiments were not successful and achieved very low scores, and as such we didn’t develop them further, or include them in this work.  If others are able to get these approaches to achieve success at DiscoveryWorld tasks, we greatly look forward to seeing them and trying their approach.
>
>
> **Opinion on whether further development should focus on large language models, or an integration with other methods such as planning and reinforcement learning:**
>
> **Response:** Thanks for this question -- much of our work (and indeed the general direction of the field) appears to focus on merging language model agents with formal planning or other symbolic formalisms, and there's been a great deal of work combining language models with reinforcement learning in various capacities.  We view all these approaches as complementary, and imagine success at DiscoveryWorld tasks is likely to require integrating many of these approaches.
>
>
> **Prior Knowledge vs Learned Knowledge:** Is it more compelling a result for a model to do well on these tasks with extensive prior knowledge, or minimal prior knowledge?
>
> **Response:** DiscoveryWorld discoveries are explicitly designed to have minimal reliance on prior knowledge, so that this benchmark can explicitly test an agent's ability to acquire critical knowledge from interacting with the environment, then apply that knowledge in the scientific discovery process.  To do this, we made the scenarios take place on a hypothetical "Planet X" in the near future, where the laws of nature may be a little different than they are on Earth.  Nominally, what is the same are the research methods for performing good science -- a scientist would still need to perform good science by collecting appropriate samples, running appropriate experiments, using appropriate analyses, and drawing appropriate conclusions.  We're agnostic on whether it's better for a model to have this kind of scientific method/process knowledge before it is tested on DiscoveryWorld, or whether it acquires that knowledge through repeated play of DiscoveryWorld -- both are interesting, and this benchmark supports both methods of operation.  More pragmatically, we're excited about any model doing well across DiscoveryWorld tasks, suggesting that we might then productively apply it more generally to (costly) real-world scenarios for automated scientific discovery with some confidence in (and characterization of) its abilities.

---

> > ### Comment · Reviewer_TdaW · 2024-08-28
> > **Thank you for the response**
> >
> > I thank the authors for response. In general, I like this work a lot and my open-ended questions are simply for some general opinions regarding the field. Though there are some missing pieces, I do understand that there is already so much that authors have done. Therefore, I keep my initial rating and would like to vote for acceptance.

---

### Official Review · Reviewer_BgaZ · 2024-07-26
**An interesting test suite for scientific discovery capabilities for LLM agents**

**Rating:** 6
**Confidence:** 5
**Correctness:** Yes, it is constructed in a sound way.
**Clarity:** Yes, it is well written.

**Review:**

Overall, I think it's an interesting benchmark for testing scientific discovery capabilities for LLM agents. The proposed benchmark does demonstrate the higher-level spirit of human scientific discovery and can benefit the future development of stronger LLM agent models. I'll discuss concrete strengths and weaknesses below.

**Strengths:**

- The paper is well-written, and the motivation to study scientific discovery, or in concrete words, the ability of hypotheses testing, design and run experiments, and draw conclusions, is good and may have broader impacts in the community.

- The tasks are well-designed, covering a wide range of tasks from multiple subjects. And the tasks look interesting! Also, it's good to see that tasks cover multiple levels of difficulties.

- The explanation check is interesting! The automatic grading part for explanatory knowledge is particularly useful.

**Additional Feedback:**

No further comments.

**Documentation:**

The code and the environment are publicly available on GitHub. No maintenance plans.

**Ethics:**

There are no foreseeable ethical concerns.

**Limitations:**

Yes, the authors have adequately addressed the limitations and potential negative societal impact of their work.

**Opportunities For Improvement:**

- The cost is high. A run for all tasks is around 3k-8k USD. The authors should consider having a subset of tasks that is easier to evaluate and can push this benchmark to a broader audience by lowering the computation & budget barrier.

- It's actually not 120 tasks. it's 8 tasks with 3 difficulty levels and 5 episodes for each task. The 120 tasks claim might be overwhelming.

- The human baseline results are not well-studied. Any comparisons with respect to the differences of explanatory knowledge between the ones made by humans and machines? And is there any task performance correlation analysis?

**Relation To Prior Work:**

Yes, it clearly discussed the relationship to prior work.

**Summary And Contributions:**

This paper introduces DiscoveryWorld, a benchmark suite designed to evaluate the scientific discovery capabilities of LLM agents. It involves a process of forming hypotheses, testing and revising hypotheses, making experiments, and drawing explanations from the observations. It includes 8 categories of tasks covering several domains, Results comparing humans and LLMs suggest that the proposed benchmark is challenging.

---

> ### Author Rebuttal · Authors · 2024-08-16
>
> We’d like to thank the reviewer for their kind and thoughtful comments, and we’re glad you enjoyed the work, and for the overall positive comments and rating. We hope to address the constructive feedback comments here.
>
> - Thanks for noting the paper is well-written, the tasks are well-designed and cover a wide range of tasks across subjects, and that the full scientific discovery cycle of hypothesis testing, designing-then-running experiments, and drawing conclusions is potentially broadly impactful.
> - Thanks also for the kind comments on our novel metrics/measurement instruments for scientific discovery (like the explanation check, and automatic grading for explanatory knowledge).
>
>
> **Responses to Questions:**
>
> **Cost:** The reviewer mentioned concern regarding the current cost of running agents in this environment, given the cost of long trajectories (e.g. 1000 steps) adds up.
>
> **Response:** We do share this concern but believe in the following budget-friendly mitigating factors:
> - *Decreasing costs:* The cost of LLM inference tends to decrease by an order-of-magnitude year-over-year, suggesting cost will not be a barrier for many in the near-term for current models.
> - *Low-resource configurations:* We provide several low-cost modes (e.g. "easy" and "unit test" modes) that have short trajectories that are much lower cost to run, even at current pricing.  We will make this more explicit in the paper.
> - *Pressure:* Inference costs become lower in response to pressure, and having novel benchmarks in the 1000-step category (like DiscoveryWorld) will provide pressure for developing new techniques (e.g. trajectory compression, macro-action sequences with minimal LLM involvement, etc.) that ultimately drive down this cost further.
>
>
> **Task Counting:** We appreciate the comment about how tasks are counted -- we use the standard convention of reporting unique task instances by unique task description prompts (e.g. MineDojo (NeurIPS 2022)) generated by procedural tasks.  We agree that it's often better just to say (N tasks * M difficulty levels * K major parametric variations) within the official benchmark, even if it's more verbose.
>
> **Further Explore Comparison of Explanatory Knowledge between Human Scientists vs Agent Scientists:** We agree, there are lots of interesting and detailed analyses that could be done by examining the explanations/notebooks of the human scientists and comparing them with the explanations provided by different agent models -- and we provide the raw notes generated by the human scientists for this purpose ( https://github.com/allenai/discoveryworld/tree/main/data ). For example, here's one of the shorter notes from a human scientist performing the Archaeology (Challenge) task, which involves validating a novel radioisotope to use for radioisotope dating from a list of 4 possible radioisotopes:
>
> ```
> TOOLS I USED WHEN SOLVING THIS SCENARIO:
> Spreadsheet: Google Sheets
> Statistics: R
>
> Notes
>
> 6 sites, 3 uncovered, find the most ancient artifact
>
> 3 artifacts
>
> The uncovered are bronze, stone, iron, stone should be the oldest as it requires less technology,
> I should find the most similar one to the stone, from all the artifacts
>
> Get data for all dig sites with the measurement tools
>
> Some measures for that are cosine similarity and dot product similarity
> I should also measure similarity for the other two tools made of bronze and iron
>
> HYPOTHESIS: Similarity measures could help me identify the oldest artifact
> SUPPORTING EVIDENCE:
> - We have 3 uncovered dig sites with identified materials iron, bronze, stone
> - We plot a similarity matrix in R
> - See which of the artifacts is more similar to the stone artifact
> - The result is the artifact from site 4, with cosine similarity to stone of 0.98
>
> Failed
> ```
>
> Performing detailed analyses on this data would likely require developing new experimental methods and coding schemes suited for this work, and performing a large number of case studies on this data.  For example, this participant makes part of the insight required for this discovery (that `stone` artifacts should be the oldest), but fails to make the follow-on inference (that `bronze` artifacts should be newer than stone, and older than `iron` artifacts -- i.e. `stone < bronze < iron`) -- or in other words, that they should be looking for a trend across 3 data points.  As such, they seem to get distracted by the format of the data (each artifact has 4 radioisotope concentration measurements, which can be thought of as a vector of these 4 different measurements) and use a vector-based analysis method (e.g. cosine similarity) when what they should really be doing is looking for a trend in one of the vector dimensions.
>
> Developing the methods and case studies to perform this analysis across all the tasks would likely require at least an additional paper worth of material and follow-on validation, and as such we leave this for future work, and provide the data in case others are interested in performing these analyses as well.

---

### Official Review · Reviewer_ub6S · 2024-07-29

**Rating:** 8
**Confidence:** 4

**Review:**

DiscoveryWorld is quite a unique and ambitious benchmark and environment for AI applications: measuring progress on scientific discovery by AI agents. The benchmark itself is quite novel, well implemented and explained, and provides a good toy environment for the scientific process. A nice addition is the inclusion of expert human evaluations, which shows the gap between current zero-shot LLM agents and humans. There is some scope for improvement to the benchmark, particularly in terms of task complexity and domain specificity. One major drawback is how prohibitively expensive it is to run the full benchmark with agents based on proprietary LLMs like GPT-4o. Overall, I think this is an excellent paper and DiscoveryWorld is going to be quite useful for agentic research for scientific discovery. I recommend a clear accept.

**Strengths:**

- The authors attempt to provide a step towards a really loft goal: automatic knowledge extraction and planning via actions in an environment. This is considered a long-shot by most experts today, even out of reach for AI. This toy benchmark will serve as an important milestone for the ability of AI agents to accomplish this goal in a small, simulated environment.

- The DiscoveryWorld environment is quite novel, and combining navigation based environments, game worlds with actions, generalizing scientific discovery tasks, as well as trying to benchmark end-to-end action, discovery, and planning for hypothesis testing

- The benchmarks make sense, and the inclusion of expert human results plus the clear gap between current zero-shot LLM reasoning agents validates prior qualitative observations on how LLM agents still lack significantly when to comes to scientific reasoning, planning, and knowledge extraction.

**Additional Feedback:**

No additional feedback

**Clarity:**

The paper is quite well written. In addition, the supplementary material (e.g. Github, website, pip package) etc. are also quite well-written and easy to use. In summary, the authors have done an excellent job at making DiscoveryWorld accessible to researchers.

**Correctness:**

The benchmark, evaluation methods and evaluation design are quite appropriate. The authors also use multiple steps, and measure for repeated successes on individual tasks to reduce spurious successes.

**Documentation:**

Between the paper, appendices, supplementary material, Github, website, and python package - there is sufficient detail on how the benchmark was constructed, how evaluations are done, how the baselines from the paper were obtained, and also how to use the benchmark for future research. The benchmark is released under the Apache 2.0 license, and the paper contains sufficient detail to reproduce the baseline results (prompts, agents used, etc.)

**Ethics:**

No major ethical concerns from DiscoveryBench itself, although there might be from the agents that are built from it in the long-term.

**Limitations:**

- A major limitation of the benchmark is how expensive it is to run the full evaluation suite. In D.4, the authors mention that running the full suite for ReAct and Plan+Execute zero shot agents costs `$3600`, while for the Hypothesizer agent it goes up to `$8400`. For anyone using DiscoveryBench to build better scientific discovery agents, this might prove to be a prohibitive cost. I don't see how anyone outside of large indsutry labs or a few well-endowed universities can easily utilize this benchmark for research.

**Opportunities For Improvement:**

- I think one major opportunity of improvement is in the task complexity. I understand that this is a hard benchmark already, and AI agents are still quite far from doing well on scientific discovery problems. However, with the progress in 'agentic' research it is not hard to imagine that such models will manage to do well on this benchmark in a couple of years. In that case, it would be helpful to have additive/recursive/compositional tasks which built upon the smaller individual tasks as they are currently proposed. This could be a part of v2 of DiscoveryBench.

- Another opportunity for improvement, which relates to the long-term usefulness of this benchmark, is connecting the dots between the current unit test themes with real steps that are often repeated in real-world science. For examplem the tasks of pick-and-give object and pick-and-place object has a direct analog to chemical experiments at the middle/high-school level. Similarly, measure an object with an instrument can be related to various sciences (e.g. physics). This sort of analogous examples will help the community improve the benchmark with more complex tasks in the future.

**Relation To Prior Work:**

Yes, the authors do a very good job at establishing the relation of DiscoveryWorld to prior environments

**Summary And Contributions:**

The authors introduce a novel environment and benchmark, DiscoveryWorld, for developing and evaluating AI agents for scientific knowledge discovery. The environment is text-based, with an optional visual overlay. There are total of 8 different areas of scientific 'themes', each with 3 level of difficulty for each task. With several parametric variations available, the 'official' eval set consists of a total of 120 tasks by considering the first five variations generated by randome seeds 0-4. The authors also include an evaluation framework that includes (a) a binary task completion metric, (b) a partial performance score card for each scientific step in the full process (c) accuracy of dicovered knowledge wrt to ground truth. Automatic evaluation is performed via carefully crafted prompts containing the ground truth and agent response by a long-context (128k) Large Language Model (GPT-4o). The authors test 3 baseline agents, which are LLMs (GPT-4o) using different zero-shot prompting based planning approaches: ReAct, Plan-And-Execute, and CLIN. A comparison against 11 expert humans on these tasks is also provided.

---

> ### Author Rebuttal · Authors · 2024-08-16
>
> We’d like to thank the reviewer for their kind and thoughtful comments, and we’re glad you enjoyed the work, and for the overall positive comments and rating.  We hope to address the constructive feedback comments here.
>
> - Thanks for all the positive comments, and in particular noting that solving DiscoveryWorld would be an important milestone for AI systems on automatic knowledge extraction, discovery, and planning.
> - Thank you for acknowledging that expert human results provide a valuable reference point, highlighting the current limitations of LLM-based reasoning agents when it comes to scientific reasoning.
>
> **Responses to Questions:**
>
> **Increased task complexity:** The reviewer notes that while the tasks are hard already, adding additional ones that are much more complex may increase the lifetime of this benchmark beyond a few years.
>
> **Response:** Thanks for this comment, and we appreciate the desire to have a benchmark with a long tail of hard tasks with many years of staying power.  One of the challenges is that many of the tasks are quite hard already, not only for agents, but also for human scientists -- we actually had to simplify a few of the original tasks for them to be solvable by a non-zero number of the human natural scientist participants (with MSc or PhD degrees) in our pool.  We're committed to maintaining the benchmark, so if agents rapidly gain performance within a year or two, we're eager to add more tasks of harder complexity -- it would just be challenging to find human scientists to norm their difficulty. We also welcome the research community to contribute and propose new tasks.
>
> **Cost:** The reviewer mentioned concern regarding the current cost of running agents in this environment, given the cost of long trajectories (e.g. 1000 steps) adds up.
>
> **Response:** We share this concern but believe in the following budget-friendly mitigating factors:
> - *Decreasing costs:* The cost of LLM inference tends to decrease by an order-of-magnitude year-over-year, suggesting cost will not be a barrier for many in the near-term for current models.
> - *Low-resource configurations:* We provide several low-cost modes (e.g. "easy" and "unit test" modes) that have short trajectories that are much lower cost to run, even at current pricing.  We will make this more explicit in the paper.
> - *Pressure:* Inference costs become lower in response to pressure, and having novel benchmarks in the 1000-step category (like DiscoveryWorld) will provide pressure for developing new techniques (e.g. trajectory compression, macro-action sequences with minimal LLM involvement, etc.) that ultimately drive down this cost further.

---

### Decision · Program_Chairs · 2024-09-26

**Decision:**

Accept (Spotlight)

**Comment:**

The paper introduces a novel environment and benchmark, DiscoveryWorld, for developing and evaluating AI agents for scientific knowledge discovery.

Summary of Strengths:

1. The problem is important and the idea is interesting.
2. The DiscoveryWorld environment is novel.
3. The benchmarks are reasonable.
4. The paper is well-written.

Summary of Weakness:
1. The cost is high.